# VIDEO-STAR: REINFORCING OPEN-VOCABULARY ACTION RECOGNITION WITH TOOLS

**Zhenlong Yuan**[1][*][†]**, Xiangyan Qu**[1][*]**, Chengxuan Qian**[1][§]**, Rui Chen**[1]**, Jing Tang**[1]**,**
**Lei Sun**[1][‡]**, Xiangxiang Chu**[1]**, Dapeng Zhang**[1]**, Yiwei Wang**[2]**, Yujun Cai**[3][§]**, Shuo Li**[4]

[1] AMAP, Alibaba Group, [2] University of California at Merced,
[3] University of Queensland, [4] Case Western Reserve University

[*] Equal contribution   [‡] Project Lead   [§] Corresponding Author

## ABSTRACT

Multimodal large language models (MLLMs) have demonstrated remarkable potential in bridging visual and textual reasoning, yet their reliance on text-centric priors often limits their ability to disentangle semantically similar actions in open-vocabulary scenarios. To address this, we propose Video-STAR, a framework that harmonizes contextual sub-motion decomposition with tool-augmented reinforcement learning for open-vocabulary action recognition (OVAR). Unlike prior methods that treat actions as monolithic entities, our approach innovatively decomposes actions into discriminative sub-motions for fine-grained matching while dynamically invoking domain-specific tools for cross-modal interleaving, thereby enabling category-specific reasoning capacity and reducing cross-modal hallucination. Moreover, by designing a hierarchical reward that balances tool-usage efficiency, sub-motion relevance, and structural coherence in reasoning, our method autonomously leverages external tools to prioritize sub-motion patterns without explicit supervision, transmitting from text-centric reasoning to visually grounded inference. Extensive evaluations on HMDB-51, UCF-101, SSv2, Kinetics-400, and Kinetics-600 datasets demonstrate our state-of-the-art performance, outperforming existing methods in distinguishing fine-grained actions and handling cross-modal hallucination, validating our excellent robustness and generalization.

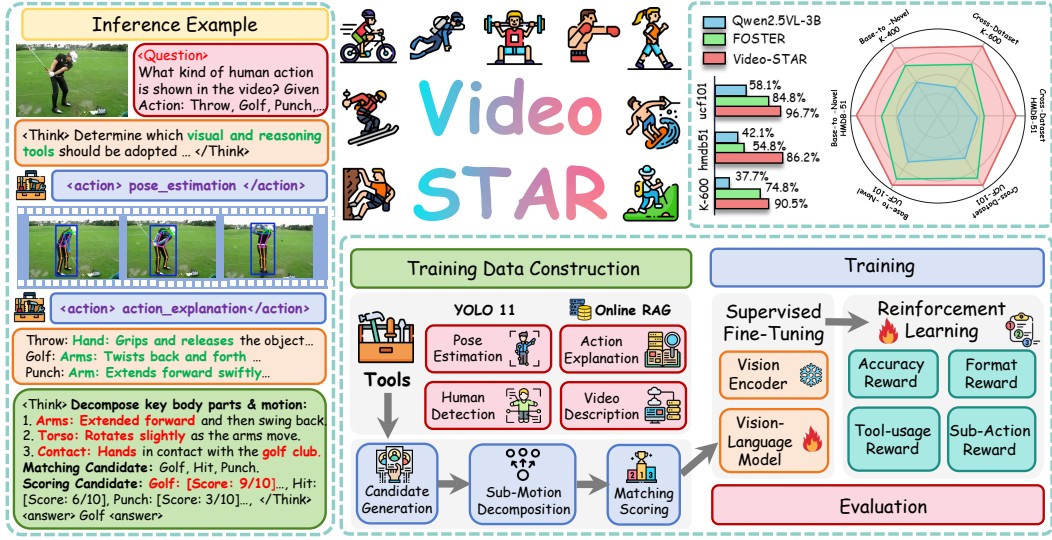

[†]Work done during the internship at AMAP, Alibaba Group.

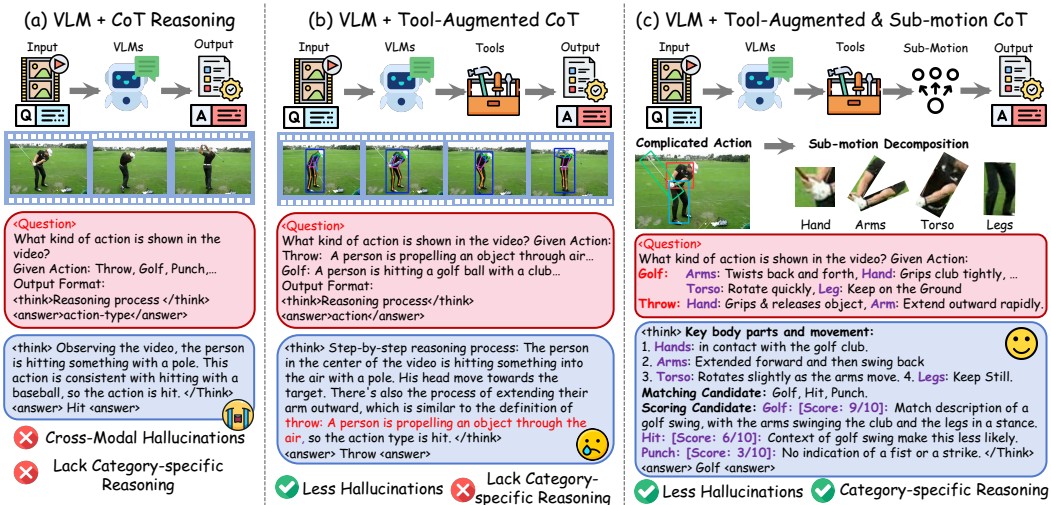

Figure 1: **Key insight of Video-STAR.** (a) MLLMs + CoT is prone to hallucinations due to over-reliance on text-centric reasoning while ignoring visual cues. (b) MLLMs + Tool-Augmented CoT mitigates hallucinations by integrating domain-specific tools to extract visual information. However, both (a) and (b) lack category-specific reasoning capabilities and struggle to distinguish semantically similar or complex actions. (c) Video-STAR enhances reasoning capacity by introducing contextual sub-motion decomposition, which disentangles actions into discriminative motion primitives. This enables fine-grained action discrimination and robust performance in open-vocabulary scenarios.

## 1 INTRODUCTION

Open-vocabulary action recognition (OVAR) aims to identify action classes not present during training, extending beyond the closed-vocabulary constraints of traditional action recognition systems (Yao et al., 2019; He et al., 2019). This capability is crucial for real-world applications where novel actions frequently emerge, such as surveillance systems, human-robot interaction, and content analysis platforms. While early approaches (Feichtenhofer et al., 2019; Lin et al., 2019) relied on hand-crafted features and limited class taxonomies, recent advances in vision-language pretraining (Radford et al., 2021; Jia et al., 2021) have opened new possibilities for zero-shot action understanding. However, OVAR remains challenging due to the inherent complexity of human actions, which often involve subtle temporal dynamics, spatial relationships, and semantic similarities that are difficult to distinguish through conventional approaches.

Recent multimodal large language models (MLLMs) have shown strong visual reasoning and cross-modal understanding (Shen et al., 2025; Feng et al., 2025). Through a unified architecture that tokenizes visual signals and fuses them with text via cross-modal attention, they achieve fine-grained grounding and stepwise reasoning. Systems such as Qwen-VL (Bai et al., 2025a) and GPT-4 (OpenAI et al., 2024) consequently deliver strong performance on image captioning (Zhao et al., 2025a), visual grounding (Bai et al., 2025b), and video understanding (Zhang et al., 2024).

Despite recent advances, these models still encounter two critical limitations when applied to OVAR tasks: ❶ Existing approaches rely heavily on text-based chain-of-thought (CoT) reasoning for visual understanding, which leads to misalignment between temporal visual features with discrete textual representations. As depicted in Fig. 1 (a), such misalignment makes the models prone to cross-modal hallucinations during reasoning. ❷ Current methods typically struggle to handle semantically similar actions in open-vocabulary scenarios. The unpredictability of actions in open-vocabulary scenarios prevents models from learning discriminative patterns directly. As shown in Fig. 1 (b), such inability in distinguishing semantically ambiguous actions may lead to erroneous classification.

To solve the above-mentioned challenges of *similar-action misclassification and cross-modal hallucinations*, we argue that effective OVAR requires two key capabilities: ❶ **Enabling Fine-grained action discrimination** through **contextual sub-motion decomposition**, which enables the model to disentangle hierarchical actions into discriminative sub-motion primitives for fine-grained classification. (e.g., "shoot basketball": "leg bending" → "torso jumping" → "arm extending" → "hand-ball

releasing"). ❷ **Enhancing cross-modal interleaving** through **multimodal CoT reasoning**, which guides the model to autonomously invoke domain-specific tools for enhanced visual-semantic representation. While recent MLLMs (Zhang et al., 2025a; Zheng et al., 2025) have demonstrated the potential of tool-augmentation in open-world video understanding (Qian et al., 2025; Su et al., 2025a), these approaches typically operate on frame-level manipulations (e.g., zooming, cropping) that lack explicit modeling of motion continuity. In contrast, OVAR requires capturing both fine-grained temporal dynamics and hierarchical sub-motion dependencies. Consequently, a unified framework that harmonizes sub-motion decomposition with tool-mediated perception to ensure both spatial precision and semantic coherence in OVAR is urgently needed.

Therefore, we introduce **VIDEO-STAR**, a framework that synergistically integrates contextual sub-motion decomposition with tool-augmented reinforcement learning (RL) for open-vocabulary action recognition. Specifically, during the inference process, our method dynamically invokes domain-specific tools like pose estimation, human detection, and online retrieval for cross-modal interleaving, thereby effectively resolving visual-semantic ambiguities and reducing hallucination. Moreover, to enable category-specific reasoning capacity, we formulate action recognition as a sequential decision-making process at the following stages: ❶ decomposing actions into discriminative sub-motion primitives, ❷ matching these primitives to several candidate actions, ❸ scoring candidates based on hierarchical relevance. Furthermore, we propose a reward mechanism that jointly optimizes for tool-usage efficiency, sub-motion relevance, and structural coherence in reasoning. This ensures that tools are activated only when they contribute meaningful insights, while sub-motion hierarchies are weighted to prioritize semantically salient components. Our contributions are threefold:

- **Tool-Augmented Agentic RL.** VIDEO-STAR introduces a novel framework that unifies sub-motion decomposition with tool-augmented reinforcement learning, thereby enabling robust open-vocabulary action recognition through dynamic, visually grounded reasoning.
- **Multi-Tool Integration.** We design a multimodal tool library integrating pose estimation, human detection, and online retrieval, which dynamically augments reasoning with domain-specific knowledge to resolve cross-modal hallucinations during reasoning.
- **Empirical Effectiveness.** Experimental results on HMDB-51, UCF-101, SSv2, Kinetics-400, and Kinetics-600 datasets demonstrate that our method achieves state-of-the-art performance, outperforming existing methods by significant margins while maintaining computational efficiency.

## 2 RELATED WORK (EXTENDED VER. IN APPX. B)

**Open-Vocabulary Action Recognition.** Recent methods for open-vocabulary action recognition (OVAR) primarily leverage CLIP's cross-modal alignment. Early approaches like Action-CLIP (Wang et al., 2021) and ViFi-CLIP (Rasheed et al., 2023) adapt CLIP via full fine-tuning or joint encoder optimization, but often overfit to static features. Parameter-efficient strategies, such as Adaptformer (Chen et al., 2022) and ST-Adapter (Pan et al., 2022), use compact modules to distill temporal knowledge, while prompt engineering (AIM (Yang et al., 2023), VPT (Ju et al., 2022)) biases models toward action semantics. However, these methods struggle with cross-architecture generalization due to limited capacity or rigid designs. Recent works like Open-VCLIP (Weng et al., 2023) and FROSTER (Huang et al., 2024b) improve generalization via weight interpolation and residual distillation, yet remain constrained by static assumptions. In contrast, Video-STAR introduces sub-motion decomposition with tool-augmented RL for robust open-vocabulary inference.

**Multimodal LLMs Reasoning.** Recent advances in large language models (LLMs) have demonstrated that RL-based post-training can significantly enhance reasoning capabilities, as exemplified by OpenAI-o1 (Jaech et al., 2024) and DeepSeek-R1 (Guo et al., 2025). These paradigms have been extended to multimodal language models (MLLMs) for tasks like mathematical VQA (Peng et al., 2025), image segmentation (Liu et al., 2025a), and video understanding (Feng et al., 2025). However, existing methods struggle with long-sequence hallucination (Chen et al., 2025) and limited cross-modal interaction. To address this, we propose multimodal CoT reasoning via tool-augmented RL, which explicitly reduces hallucination through dynamic tool integration.

**Tool-Augmented Agentic System.** Recent advancements in LLMs have shown that external tools can enhance multimodal reasoning. Early works like FAST (Sun et al., 2025) and MVoT (Li et al., 2025a) introduce visual evidence into reasoning, forming multimodal CoT for image tasks. LLaVa-Plus (Liu et al., 2023b) pioneered training strategies for tool use, while VPD (Hu et al., 2024)

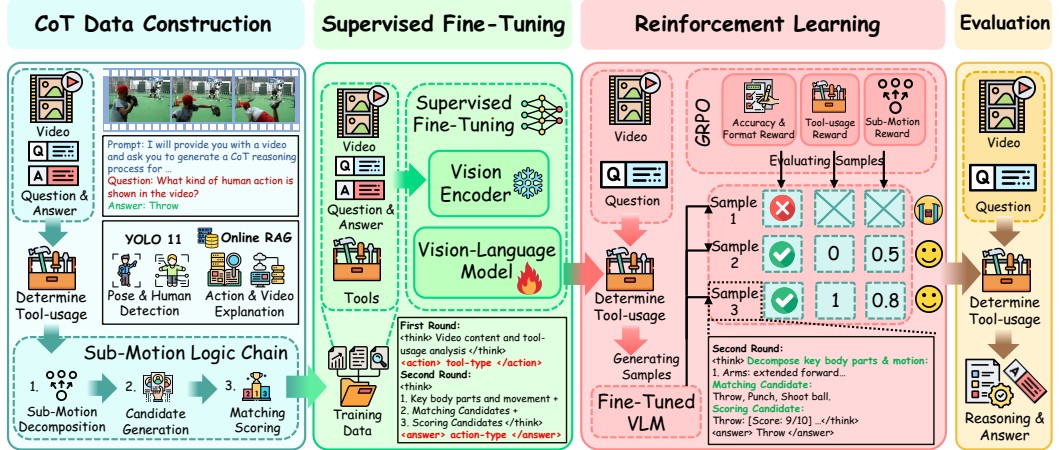

Figure 2: **Pipeline of Video-STAR.** (i) Introduce a three-stage sub-motion logic chain to construct tool-augmented reasoning data that decomposes actions into discriminative sub-motions. (ii) Pre-train the MLLMs on structured reasoning chains and fine-tune it for domain-specific adaptation. (iii) Adopt the GRPO algorithm for reinforcement learning, which optimizes a hierarchical reward function considering both tool-usage and sub-motion to ensure robust and consistent inference.

leveraged program-derived data to transfer tool skills. Recent methods like TACO (Ma et al., 2024) and PyVision (Zhao et al., 2025b) expand tool use with RL. However, existing approaches rely on static tool pipelines, limiting their ability to adapt to complex, open-vocabulary actions. Our framework addresses this by explicitly modeling the contextual sub-motions and dynamically balancing tool usage efficiency with hierarchical motion relevance through a structured reward mechanism.

## 3 METHODOLOGY

**Overview.** We propose Video-STAR, a unified framework that synergizes contextual sub-motion decomposition with tool-augmented reinforcement learning, as illustrated in Fig. 2. Sec 3.1 first details the problem formulation of the inference process and the tool library. Sec 3.2 details the **Training Data Construction** through multimodal CoT generation, synthesizing video-query pairs with predefined prompts. Sec 3.3 presents **Agentic Supervised Fine-tuning**, where the Qwen2.5-VL base model is pretrained on the reasoning data for domain-specific adaptation. Sec 3.4 outlines **Agentic Reinforcement Learning**, which introduces a hierarchical reward function that balances tool-usage, sub-motion, and structural coherence for inference. The above-mentioned three stages are tightly coupled: the training data provides the foundation for SFT, which in turn initializes the RL policy, ensuring end-to-end alignment between perception, reasoning, and action inference.

### 3.1 PROBLEM FORMULATION.

In open-vocabulary action recognition (OVAR), the task is to identify action classes not seen during training. The **input consists of a video** $V$ **and a query** $Q$, while the **output is a predicted action** $A$ that may belong to unseen categories during training. To achieve this goal, we design the following two-stage framework that dynamically integrates domain-specific tools: ❶ The first stage selects the most relevant tool from an augmented toolset $T = \{T_p, T_d, T_a, T_v\}$. ❷ The second stage combines tool outputs with raw inputs for final classification. This architecture ensures contextual consistency between visual observations and semantic reasoning by explicitly modeling the interaction between perceptual features and task-specific knowledge. The implementation details are as follows:

**Stage 1: Tool Selection.** Given a video $V$ and query $Q$, the model analyzes contextual cues to determine which tool is most relevant for resolving our task. The toolset contains four components: $T_p$ for pose estimation, $T_d$ for human detection, $T_a$ for action explanation, $T_v$ for video description. The first-stage output $y' \sim \pi_\theta(\cdot|V, Q; T)$ produces a selected tool call $C \subseteq T$ along with intermediate results $R$, where $\pi_\theta$ represents the policy network of MLLMs parameterized by weights $\theta$.

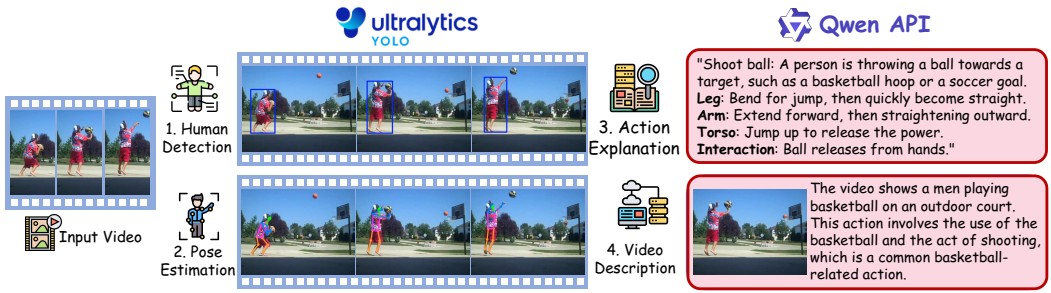

Figure 3: **Tool Libirary.** Given the input video, Video-STAR respectively adopts the YOLO 11 for human detection & pose estimation, and the Qwen API for action explanation & video description.

**Stage 2: Result Integration & Prediction** The second stage refines action recognition by integrating tool outputs $R$ with original inputs. For the visual tools $T_p$ and $T_d$, extracted features $F$ are concatenated with raw video frames $V$ to enrich spatial-temporal representations. For the semantic reasoning tool $T_a$ and $T_v$, the generated explanation $E$ is appended to the original query $Q$ to provide contextual grounding. The final prediction $A$ is computed as:

$$A \sim \pi_\theta(\cdot | V \oplus F, Q \oplus E; T), \tag{1}$$

where $\oplus$ denotes feature concatenation for visual modalities and text appending for reasoning, while $\pi_\theta$ maintains the same policy network of MLLMs parameterized by weights $\theta$.

**Tool Library** As shown in Fig. 3, our toolkit library contains four sub-tools: ❶ **Human Detection**: we adopt the YOLO 11 (Khanam & Hussain, 2024), which employs a hybrid spatial attention mechanism for anchor-free human detection. ❷ **Pose Estimation**: we introduce YOLO 11's 17-keypoint skeletonization capability (e.g., COCO format) to capture sub-pixel joint positions for fine-grained motion analysis. ❸ **Action Explanation**: we leverage the Qwen API for *online retrieval augmentation (RAG)*, then redefine actions through transitional phases (e.g., "stand" → "transit from sit or lay to stand"). ❹ **Video Description**: we extract temporally salient frames, then adopt the Qwen API for online RAG to disambiguate multi-phase action sequences (e.g., "sit-stand-walk") via spatial-temporal descriptions. The detailed descriptions are shown in Appendix D.

### 3.2 TRAINING DATA CONSTRUCTION

Existing MLLMs methods typically suffer from two critical limitations when applied to OVAR tasks: ❶ **The over-reliance on textual priors** that neglects domain-specific visual cues, and ❷ **The inability to distinguish ambiguous actions** in open-vocabulary scenarios. To address these issues, we construct a novel training dataset by integrating multimodal CoT reasoning with tool-augmented decomposition, thereby ensuring both spatial-temporal alignment and semantic accuracy.

**Data Collection.** As shown in the cyan part of Fig. 2, our dataset is constructed via a three-stage logic chain. First, we curated an initial pool of approximately 7,000 video-query pairs from the HMDB-51 dataset (Kuehne et al., 2011). We then leveraged Qwen2.5-VL-72B (Bai et al., 2025a) to synthesize initial chain-of-thought (CoT) reasoning sequences for these pairs. Crucially, to ensure data quality, we introduced a rigorous filtering stage: each generated reasoning chain was scored and reviewed by the closed-source Qwen-VL-Max model as an expert validator. Chains containing factual errors or logical inconsistencies(e.g., "hallucinated" or incoherent reasoning) were systematically discarded. This validation-and-pruning procedure yielded a final set of approximately 5,000 high-fidelity reasoning chains, specifically curated for supervised fine-tuning (SFT). These chains are designed to dynamically integrate visual features and domain-specific knowledge, thereby bridging the gap between text-centric reasoning and visually grounded inference.

**Generating Pipeline.** The core innovation lies in our three-stage framework for action recognition:

- **Sub-motion Decomposition** first breaks down the holistic action into discriminative sub-motion primitives by identifying key body-part interactions. For "shoot ball" is decomposed into "body bending → leg extending → feet-ball interaction". This decomposition addresses the inherent ambiguity of holistic perception by isolating perceptually distinct motion patterns.
- **Candidate Selection** then maps each sub-motion to pre-defined action definitions, generating 2-3 semantically relevant candidates. In our example, "body bending" suggests "run", "leg extending"

implies "hurdle", and "feet-ball interaction" aligns with "shoot". This step narrows the search space while preserving semantic diversity, ensuring robust open-vocabulary generalization.

- **Matching Scoring** finalizes recognition by comparing all sub-motions against each candidate's detailed definition. The system evaluates spatial precision and description alignment, ultimately selecting the highest-scoring candidate. In our case, the combination of precise spatial interactions and semantic coherence with the definition "shoot ball" would lead to the correct classification.

**Data Assessment** To ensure the reliability of the constructed data, we introduce a specialized LLM to evaluate its factual accuracy and logical coherence. This model systematically removes entries containing inconsistent reasoning steps or unverified claims, resulting in a refined dataset that combines explicit tool usage with logically coherent, evidence-supported reasoning.

### 3.3 AGENTIC SUPERVISED FINE-TUNING

**After constructing the training data**, we introduce a **cold-start phase** that prioritizes supervised fine-tuning (SFT) before applying reinforcement learning (RL). Inspiring from R1-Zero (Guo et al., 2025), we initially attempt direct RL optimization to train our method. However, preliminary experiments reveal a progressive decline in the frequency of tool invocation during policy rollouts. This behavior likely arises from a distributional discrepancy in target domain between tool-enhanced visual features and model's pretraining data. Therefore, we opt to perform SFT before RL.

Specifically, We formalize each training instance as $\mathcal{T} = (\mathcal{X}, \mathcal{I}, \mathcal{S}, \mathcal{Y})$, where $\mathcal{X}$ denotes the input modality, $\mathcal{I}$ represents task instructions, $\mathcal{S} = \{s_1, \ldots, s_T\}$ captures reasoning steps, and $\mathcal{Y}$ is the target output. The objective minimizes the negative log-likelihood of the reasoning process:

$$\mathcal{L}_{\text{SFT}} = -\mathbb{E}_{\mathcal{T} \sim \mathcal{D}} \left[ \sum_{t=1}^{T} \log p_\theta(s_t \mid \mathcal{X}, \mathcal{I}, s_{<t}) \right], \tag{2}$$

where $p_\theta(\cdot)$ is the model's conditional probability distribution. This formulation enables model to generate reasoning chains that align with both task instructions and ground-truth during pre-training.

### 3.4 AGENTIC REINFORCEMENT LEARNING

**Group Relative Policy Optimization (GRPO).** We adopt the GRPO algorithm (Jiang et al., 2025) for policy optimization. GRPO innovatively employs a groupwise comparison framework to evaluate candidate responses. Specifically, for each query $q$ paired with its ground-truth solution $a$ from dataset $D$, the algorithm generates a set of rollout trajectories $\{o_1, o_2, \ldots, o_G\}$ based on the previous policy $\pi_{\theta_{\text{old}}}$. The policy $\pi_\theta$ is then refined through the optimization of this objective function:

$$\mathcal{L}_{GRPO}(\theta) = -\mathbb{E}[q \sim P(Q), \{o_i\}_{i=1}^{G} \pi_{\theta old}(O|q)] \frac{1}{G} \sum_{i=1}^{G}$$

$$\left( min\left( \frac{\pi_\theta(o_i|q)}{\pi_{\theta old}(o_i|q)} * Adv_i, clip\left( \frac{\pi_\theta(o_i|q)}{\pi_{\theta old}(o_i|q)}, 1-\epsilon, 1+\epsilon \right) * Adv_i \right) - \beta \mathbb{D}_{KL}(\pi_\theta || \pi_{ref}) \right), \tag{3}$$

$$\mathbb{D}_{KL}(\pi_\theta || \pi_{ref}) = \frac{\pi_{ref}(o_i|q)}{\pi_\theta(o_i|q)} - log\frac{\pi_{ref}(o_i|q)}{\pi_\theta(o_i|q)} - 1, \tag{4}$$

where $\beta$ is adopted to balance the trade-off between exploration and stability during optimization. Then the advantage estimator $A_i$ is calculated using normalized rewards from the trajectory group:

$$A_i = \frac{r_i - \text{mean}(\{r_1, r_2, \ldots, r_G\})}{\text{std}(\{r_1, r_2, \ldots, r_G\})}. \tag{5}$$

Each trajectory $o_i$ receives a binary reward $r_i \in \{0, 1\}$ through a rule-based verification system designed to mitigate reward manipulation risks.

**Reward Design.** Effective reward functions should balance accuracy, structural coherence, tool-use efficiency, and sub-motion relevance to support the training of agentic systems for OVAR tasks. While traditional reward design primarily prioritizes the correctness of the answer and format,

we extend this by incorporating hierarchical sub-motion relevance and tool-mediated perception to guide structured reasoning over sequential sub-motions.

The total reward integrates four components: accuracy reward $R_{\text{acc}}$, format reward $R_{\text{format}}$, tool-usage reward $R_{\text{tool}}$, and sub-motion reward $R_{\text{sub}}$. The accuracy reward $R_{\text{acc}}$ evaluates the correctness of the final action classification, while the formatting reward $R_{\text{format}}$ penalizes unstructured or incomplete reasoning chains. The tool-usage reward $R_{\text{tool}}$ is activated only when correct answers are produced alongside valid tool invocations. The sub-motion reward $R_{\text{sub}}$ employs a hierarchical weighting mechanism to prioritize semantically salient sub-motions. Specifically, the model first generates $n$ sub-motions through contextual analysis, ordered by their relevance to the action definition. For the $k$-th ranked candidate, its weight is assigned as: $w_k = n - k + 1$. When the predicted classification matches a subset of $m$ sub-motions $\{k_1, \ldots, k_m\}$, we define: $R_{\text{sub}} = \sum_{i=1}^{m} w_{k_i} / \sum_{i=1}^{n} w_{k_i}$. This formulation ensures higher-priority sub-motions receive exponentially greater weighting, while lower-priority ones contribute proportionally less. Formally, the total reward is defined as:

$$R(\tau) = R_{\text{acc}}(\tau) + R_{\text{format}}(\tau) + \mathbb{I}_{R_{\text{acc}}(\tau)>0} \cdot (R_{\text{tool}}(\tau) + R_{\text{sub}}(\tau)), \qquad (6)$$

where $\mathbb{I}_{R_{\text{acc}}(\tau)>0}$ is the indicator function which assigns a binary value of 1 when $R_{\text{acc}}(\tau) > 0$ and 0 otherwise. This conditional structure enforces two key principles: ❶ perception-aware reasoning requires simultaneous activation of tool usage and sub-motion prioritization only when the final answer is correct, and ❷ redundant tool invocations or irrelevant sub-motions receive no reward. By coupling tool-mediated perception with hierarchical sub-motion weighting, the framework ensures that agents invoke tools meaningfully while maintaining coherent reasoning chains.

## 4 EXPERIMENT

**Experimental Settings.** We adopt two experimental settings: ❶ *base-to-novel*: For each dataset, the label space is partitioned into two disjoint subsets, *i.e.*, the base classes $Y_B$ and the novel classes $\mathcal{Y}_N$, where $\mathcal{Y}_B \cap \mathcal{Y}_N = \emptyset$. We train the models on data from $\mathcal{Y}_B$ and evaluate on testing samples from $\mathcal{Y}_B \cup \mathcal{Y}_N$. ❷ *cross-dataset*: In this setting, the model is trained on a source dataset with the label set as $\mathcal{Y}_S$ and evaluated on a target dataset with another label set as $\mathcal{Y}_T$, where $|\mathcal{Y}_S \cup \mathcal{Y}_T| \geq |\mathcal{Y}_S \cap \mathcal{Y}_T|$.

**Dataset and Metrics.** We evaluate our method on five standard action recognition benchmarks: UCF-101 (Soomro et al., 2012), HMDB-51 (Kuehne et al., 2011), Kinetics-400 (K-400) (Carreira & Zisserman, 2017), Kinetics-600 (K-600) (Carreira et al., 2018), and Something-to-Something V2 (SSv2). UCF-101 has 13,320 video clips from 101 classes, while HMDB-51 includes 6,849 videos across 51 classes. For large-scale evaluation, we use K-400 and K-600, which contain 400 and 600 classes, respectively. SSv2 contains 174 fine-grained classes. Following prior work (Rasheed et al., 2023; Ni et al., 2022; Weng et al., 2023), we report average top-1 accuracy under the two settings.

**Implementation Details.** We implement experiments on both Qwen2.5-VL-3B and Qwen2.5-VL-7B models. In stage one, we select 5,000 training samples constructed from Sec 3.2 for SFT. These same training samples are then reused for subsequent RL optimization during the stage two. The training was conducted on eight NVIDIA H20 GPUs (90 GB memory each) using a 5k sample dataset with a batch size of 8. The process took 20 hours for 600 iterations (1 epoch), with 4 rollouts per sample and a learning rate of 5e-7. More experimental details are shown in Appendix A.1.

### 4.1 MAIN RESULTS

**Base-to-novel**. In Tab. 1, we compare our Video-STAR with the state-of-the-art (SOTA) methods under the base-to-novel setting. Notably, our model is fine-tuned only on the base set of HMDB-51 and evaluated in a zero-shot manner on K-400, UCF-101, and SSv2. In contrast, all other methods are fine-tuned on the base set of each respective dataset, making our evaluation more challenging. From the results, we highlight four findings: ❶ Our Video-STAR established a new SOTA, consistently outperforming previous methods across all datasets. Specifically, Video-STAR-7B achieves **26.3%** in K-400 and **27.0%** in HMDB-51 absolute improvement over the previous SOTA. ❷ Video-STAR demonstrates strong generalization to novel categories. We see the performance gap of Video-STAR between the base and novel sets is nearly closed. Remarkably, on K-400 and UCF-101, the accuracy of novel classes exceeds that of base ones. ❸ Scaling helps the complex dataset. We observe a **10.5%** and **3.2%** absolute increase in HM on K-400 and SSv2 datasets when scaling from

Table 1: Performance comparison (Top1-Acc (%)) with the CLIP-based methods using ViT-B/16 under **the base-to-novel setting**. "HM" denotes the harmonic mean of the accuracy from the base and novel sets. Note that the best and second best performances are highlighted.

| Method | K-400 | | | HMDB-51 | | | UCF-101 | | | SSv2 | | |
|---|---|---|---|---|---|---|---|---|---|---|---|---|
| | Base | Novel | HM | Base | Novel | HM | Base | Novel | HM | Base | Novel | HM |
| ActionCLIP | 61.0 | 46.2 | 52.6 | 69.1 | 37.3 | 48.5 | 90.1 | 58.1 | 70.7 | 13.3 | 10.1 | 11.5 |
| X-CLIP | 74.1 | 56.4 | 64.0 | 69.4 | 45.5 | 55.0 | 89.9 | 58.9 | 71.2 | 8.5 | 6.6 | 7.4 |
| VPT | 69.7 | 37.6 | 48.8 | 46.2 | 16.0 | 23.8 | 90.5 | 40.4 | 55.8 | 8.3 | 5.3 | 6.4 |
| ST-Adapter | 73.6 | 62.0 | 67.3 | 65.3 | 48.9 | 55.9 | 85.5 | 76.8 | 80.9 | 9.3 | 8.4 | 8.8 |
| ViFi-CLIP | 76.4 | 61.1 | 67.9 | 73.8 | 53.3 | 61.9 | 92.9 | 67.7 | 78.3 | 16.2 | 12.1 | 13.9 |
| FROSTER | 77.8 | 64.3 | 70.4 | 74.1 | 58.0 | 65.1 | 95.3 | 80.0 | 87.0 | 18.3 | 12.2 | 14.6 |
| Open-MeDe | 77.2 | 63.8 | 69.9 | 73.6 | 56.4 | 63.9 | 94.9 | 78.5 | 85.9 | 17.1 | 12.3 | 14.3 |
| VTD-CLIP | 78.5 | 63.5 | 70.1 | 78.4 | 63.5 | 70.0 | 95.5 | 73.7 | 83.2 | 17.8 | 13.9 | 15.4 |
| AP-CLIP | 77.2 | 64.1 | 70.0 | 74.6 | 55.9 | 63.9 | 94.8 | 77.0 | 84.8 | 16.3 | 12.9 | 14.4 |
| Qwen2.5-VL-3B | 48.3 | 40.5 | 44.1 | 54.0 | 40.8 | 46.5 | 71.4 | 62.4 | 66.6 | 3.5 | 3.2 | 3.3 |
| Qwen2.5-VL-7B | 87.8 | 84.8 | 86.3 | 41.7 | 50.3 | 45.6 | 85.0 | 82.4 | 83.7 | 14.0 | 9.9 | 11.6 |
| Video-STAR-3B | 86.0 | 86.4 | 86.2 | 92.1 | 91.7 | 91.9 | 96.9 | 98.9 | 97.9 | 13.5 | 11.3 | 12.3 |
| Video-STAR-7B | 96.3 | 97.2 | 96.7 | 92.3 | 91.9 | 92.1 | 99.6 | 99.8 | 99.7 | 19.2 | 13.0 | 15.5 |

Table 2: Performance comparison (Top1-Acc (%)) with the CLIP-based methods under **the cross-dataset setting**. "Full" denotes the full validation set, while "Split" denotes evaluating across three validation splits. Note that the best and second best performances are highlighted.

| Method | UCF-101 | | HMDB-51 | | K-600 |
|---|---|---|---|---|---|
| | Full ↑ | Split ↑ | Full ↑ | Split ↑ | Split ↑ |
| ActionCLIP (Wang et al., 2021) | 77.4 | 77.5±0.8 | 48.0 | 48.2±1.5 | 62.5±1.2 |
| X-CLIP (Ni et al., 2022) | - | 72.0±2.3 | - | 44.6±5.2 | 65.2±0.4 |
| ST-Adapter (Pan et al., 2022) | 77.9 | 77.6±0.7 | 50.3 | 51.1±0.6 | 60.2±1.8 |
| Vita-CLIP (Wasim et al., 2023) | - | 75.0±0.6 | - | 48.6±0.6 | 67.4±0.5 |
| ViFi-CLIP (Rasheed et al., 2023) | - | 76.8±0.7 | - | 51.3±0.6 | 71.2±1.0 |
| Open-VCLIP (Weng et al., 2023) | 83.5 | 83.4±1.2 | 53.2 | 53.9±1.2 | 73.0±0.8 |
| FROSTER (Huang et al., 2024b) | 85.0 | 84.8±1.1 | 54.5 | 54.8±1.3 | 74.8±0.9 |
| Open-MeDe (Yu et al., 2025a) | - | 83.7±1.3 | - | 54.6±1.1 | 73.7±0.9 |
| STDD (Yu et al., 2025b) | - | 85.2±1.2 | - | 55.9±0.2 | 75.1±0.7 |
| Qwen2.5-VL-3B | 58.3 | 58.1±0.2 | 39.2 | 42.1±0.3 | 37.7±1.8 |
| Qwen2.5-VL-7B | 77.5 | 77.2±0.6 | 50.6 | 53.1±0.1 | 68.0±1.5 |
| Video-STAR-3B | 96.8 | 96.7±0.3 | 83.5 | 86.2±0.2 | 90.5±0.7 |
| Video-STAR-7B | 99.4 | 99.4±0.2 | 90.1 | 92.5±0.1 | 98.2±0.1 |

3B to 7B. However, on HMDB-51 and UCF-101, performance is already near-perfect. ❹ Generic MLLMs are insufficient. Although Qwen2.5-VL-7B is competitive on K-400 (HM 86.3), it collapses on HMDB-51 (HM 45.6) and SSv2 (HM 11.6), whereas Video-STAR maintains uniformly high performance, suggesting that our specific alignment is crucial for robustness across datasets.

**Cross-dataset**. In Tab. 2, we compare our method with the previous methods under the cross-dataset setting. We see two key observations. ❶ Video-STAR exhibits remarkable stability, achieving comparable results on the full validation sets (UCF-101*, HMDB-51*) and the splits (UCF-101, HMDB-51). This indicates that our model achieves robustness through tool usage and sub-motion decomposition. ❷ Our smaller 3B model outperforms previous state-of-the-art methods, as well as the larger Qwen2.5-VL-7B. This further demonstrates the effectiveness of our Video-STAR. Moreover, Fig. 4 presents a comparative case study. More case studies are provided in Appendix A.3.

## 4.2 ABLATION STUDIES

**Training Stages**. In rows *(a)-(d)* of Tab. 3, we compare zero-shot, full model, SFT-only, and RL-only training paradigms, respectively. The zero-shot baseline in row *(a)* yields the lowest results among all methods. SFT-only training in row *(c)* significantly improves accuracy, outperforming RL-only training in row *(d)*. We attribute the gap to the difficulty of RL in directly navigating

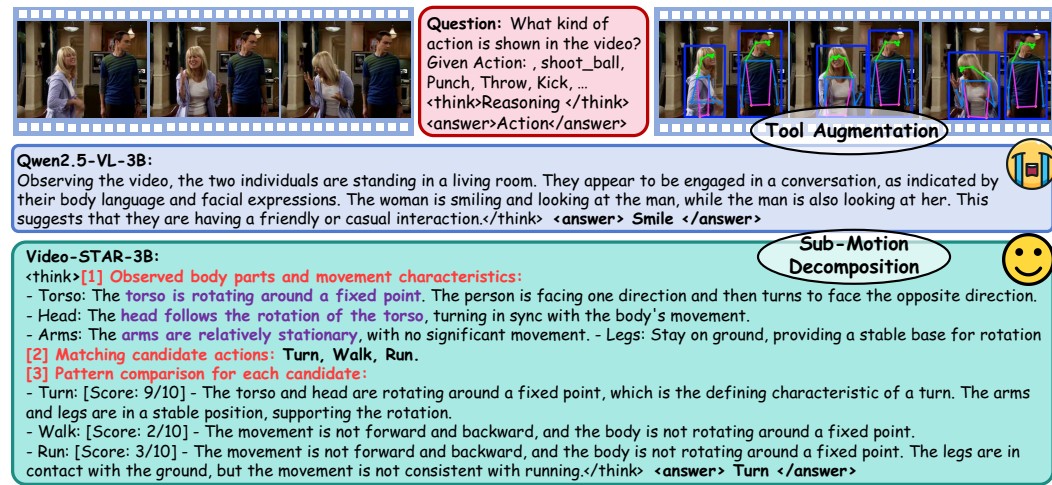

Figure 4: **Case Study between Qwen2.5-VL-3B and our Video-STAR-3B.** Qwen2.5-VL-3B misclassifies action "turn" as "smile", while our Video-STAR-3B accurately identifies the correct action.

Table 3: Ablation of key components in Video-STAR under **cross-dataset setting**. "Split" denotes evaluating across three validation splits. The best and second best performances are highlighted.

| Method | UCF-101 | HMDB-51 | K-600 | RL | UCF-101 | HMDB-51 | K-600 |
|---|---|---|---|---|---|---|---|
| | Split↑ | Split↑ | Split↑ | | Split↑ | Split↑ | Split↑ |
| (a) Qwen2.5-VL-3B | 58.1±0.2 | 42.1±0.3 | 37.7±1.8 | (g) w/. Base | 93.8±0.4 | 81.7±0.2 | 84.7±1.2 |
| (b) Video-STAR-3B | 96.7±0.3 | 86.2±0.2 | 90.5±0.7 | (h) w/o. Tol | 95.1±0.3 | 83.2±0.2 | 86.7±0.9 |
| (c) w/o. RL | 76.8±0.8 | 63.5±0.2 | 61.3±2.0 | (i) w/o. Sub | 95.7±0.3 | 84.1±0.2 | 87.9±0.8 |
| (d) w/o. SFT | 71.4±0.9 | 57.6±0.2 | 54.8±2.1 | (j) Num = 2 | 96.4±0.2 | 85.0±0.2 | 89.3±0.3 |
| (e) w/o. TOL | 87.1±0.5 | 74.9±0.3 | 78.4±1.5 | (k) Num = 4 | 96.7±0.3 | 86.2±0.2 | 90.5±0.7 |
| (f) w/o. SUB | 88.5±0.5 | 76.8±0.3 | 81.2±1.3 | (l) Num = 6 | 96.9±0.3 | 86.9±0.2 | 91.0±0.9 |

the high-dimensional reasoning space without prior supervised guidance. These results support our two-stage strategy, where SFT provides a structured reasoning foundation before RL optimization.

**Tool-Usage & Sub-Motion**. In rows *(e)-(f)* of Tab. 3, we show the effect of removing tool-usage and sub-motion decomposition from our full framework. Removing tool usage in row *(e)* causes a notable performance degradation with at least **9.1%**. This confirms the critical role of external tools in resolving cross-modal ambiguities and reducing hallucination. Similarly, removing sub-motion decomposition in row *(f)* reduces accuracy **8.2%**, **9.4%**, **9,3%** in UCF-101, HMDB-51, and K-600, respectively. This indicates the importance of breaking complex actions into discriminative primitives and enabling category-specific reasoning. The full model (row *(b)* achieves the highest scores, demonstrating that tool-augmented reasoning and sub-motion hierarchy are complementary and jointly essential for robust open-vocabulary action recognition.

**Reinforcement Learning**. In rows *(g)-(i)* of Tab. 3, we analyze the effect of different reward configurations in RL training. Removing both tool-usage and sub-motion rewards *(g)* produces the lowest performance, showing that accuracy and format rewards alone are insufficient to guide reasoning. Including sub-motion reward in row *(h)* improves results at least **1.3%**, illustrating that hierarchical motion decomposition provides strong semantic cues even without tool-mediated perception. Conversely, including tool-usage reward *(i)* also yields better performance than *(g)* with at least **1.9%**, confirming that tool integration helps resolve visual-semantic ambiguities. The full model in row *(b)*, which utilizes the complete reward function, achieves the highest accuracy. This demonstrates that our reward design is crucial for effective RL optimization in our framework.

**Number of Generation**. In rows *(j)-(l)* of Tab. 3, we analyze the impact of different numbers of generations during the RL training (GRPO). Increasing the number of candidate responses from 2 to 6 consistently improves performance across all datasets. Specifically, when generating six responses per input *Num = 6*, the model achieves the highest performance with average accuracy scores of

**96.9%**, **86.9%**, and **91.0%** on UCF-101, HMDB-51, and K-600, respectively, outperforming the configurations with *Num = 2* and *Num = 4*. However, the marginal gains diminish beyond *Num = 6*, indicating a trade-off between computational cost and performance improvement. Therefore, we opt for *Num = 4* to balance accuracy and efficiency. More ablation study is shown in Appendix A.2.

Table 4: Comparison between our agentic system and a static pipeline that adopts all tools.

| Method | UCF Acc↑ | Infer Time↓ | Tool Time↓ | Total Time↓ |
|---|---|---|---|---|
| All Tools (Static) | 97.2% | 1.86s | 2.24s | 4.10s |
| **Video-STAR 3B** | 96.7% | 1.75s | 1.43s | 3.18s |

**Agentic System vs. Static Pipeline.** To validate the effectiveness of our dynamic tool-selection policy, we compare our agentic system against a static pipeline that invariably invokes all tools. As shown in Table 4, our agentic model achieves an excellent trade-off between performance and efficiency. For a marginal 0.5% drop in accuracy on UCF-101, our approach reduces total inference time by approximately 22% (from 4.10s to 3.18s). This result demonstrates that the learned policy is adept at pruning redundant tool calls, securing substantial efficiency gains while maintaining top-tier accuracy, thus confirming the superiority of an agentic design over a rigid, static pipeline.

Table 5: Computational overhead analysis. Video-STAR achieves a massive accuracy boost with a moderate increase in latency and GFLOPs, highlighting a favorable performance-cost trade-off.

| Method | Memory↓ | Latency↓ | UCF Acc↑ | GFLOPs↓ | BSZ |
|---|---|---|---|---|---|
| Qwen-2.5VL-3B | 60.03 GB | 0.79s + 0s | 58.1% | 18,296 | 50 |
| **Video-STAR-3B** | 60.89 GB | 1.75s + 1.43s | 96.7% | 25,773 | 50 |

**Computational Overhead.** We analyzed the computational cost of Video-STAR compared to its base model, Qwen-2.5VL-3B. As reported in Table 5, the majority of the added latency comes from tool invocation (1.43s), which is spent executing external models and APIs. Despite this overhead, our framework boosts accuracy on UCF-101 from a mere 58.1% to an exceptional 96.7%. This massive +38.6% absolute improvement for a moderate increase in computational cost highlights a highly favorable trade-off and underscores the value of our tool-augmented reasoning approach.

Table 6: Ablation on tool selection, demonstrating the framework's modularity. Performance remains high when swapping core tools, proving the agentic logic is the key contribution.

| Tool Stack (3B) | UCF Acc↑ | Latency (per sample)↓ |
|---|---|---|
| YOLO 11 + Qwen (Ours) | 96.9% | 3.18s |
| YOLO 11 → OpenPose | 96.1% | 4.61s |
| Qwen → Gemini-1.5-Pro | 97.4% | 3.47s |

**Robustness to Tool Selection.** To demonstrate the framework's modularity and robustness, we conducted experiments by swapping our default tools with alternatives. As detailed in Table 6, replacing YOLO 11 with OpenPose for pose estimation or substituting the Qwen API with Gemini-1.5-Pro for semantic reasoning results in consistently high performance. Although latency varies with the tool choice (e.g., OpenPose is slower, Gemini-1.5-Pro is more powerful but more expensive), the accuracy remains robust. This proves that our primary innovation lies in the underlying agentic logic, which can flexibly orchestrate a variety of tools, rather than on a single, specific tool stack.

## 5 CONCLUSION

In this work, we propose Video-STAR, a novel framework that integrates contextual sub-motion decomposition with tool-augmented reinforcement learning. By decomposing complex actions into discriminative sub-motions and leveraging domain-specific tools for spatial-temporal alignment, our approach bridges the gap between text-centric reasoning and visually grounded inference. The hierarchical reward mechanism ensures efficient tool usage while prioritizing semantically salient sub-motion patterns, enabling robust performance in various scenarios. Extensive experiments demonstrate our state-of-the-art performance, outperforming existing methods by significant margins. Future work will explore cross-modal generalization to broader video understanding applications.

## ETHICS STATEMENT

Our work centers on advancing open-vocabulary action recognition through the Video-STAR framework, which leverages publicly available video datasets (e.g., HMDB-51, UCF-101, Kinetics) and domain-specific tools (e.g., YOLO for pose estimation, Qwen API for semantic reasoning). All training and evaluation data were sourced from existing benchmarks with established ethical guidelines, ensuring compliance with data usage policies and minimizing risks of privacy violations. The framework processes visual and textual inputs without extracting or retaining personally identifiable information (PII), focusing solely on abstract action patterns and contextual sub-motions to avoid unintended individual tracking. We emphasize the responsible deployment of our model, advocating for its use in applications that prioritize human well-being, such as assistive technologies or sports analytics, while explicitly discouraging misuse in surveillance or discriminatory practices. To promote transparency, we commit to open-sourcing our code, detailed methodologies, and anonymized training data, enabling independent verification and fostering community-driven ethical oversight. Our experiments adhered to rigorous academic standards, and we remain dedicated to addressing potential biases in tool integration or sub-motion decomposition to ensure equitable performance across diverse populations and scenarios.

## REPRODUCIBILITY STATEMENT

To ensure the full reproducibility of our findings, we provide comprehensive implementation details in the paper and its appendices. The construction of the training data, including the three-stage sub-motion logic chain and multimodal chain-of-thought (CoT) generation process, is detailed in Sec 3.2 and Appendix E. The architecture of the Video-STAR framework, encompassing contextual sub-motion decomposition, tool selection mechanisms, and the hierarchical reward function, is described in Sec 3.3 and Sec 3.4. Implementation specifics for the GRPO algorithm, including reward function components (accuracy, format, tool-usage, and sub-motion rewards) and training hyperparameters (e.g., learning rate, batch size, and generation settings), are presented in Sec 3.4 and Appendix A.1. The tool library, including YOLO 11 for human detection and pose estimation, as well as the Qwen API for action explanation and video description, is fully documented in Appendix D. In line with our commitment to open science, the source code, pre-trained models, and detailed documentation will be made publicly available.

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

## APPENDIX

In this appendix, we provide more experimental details, related work, tool library, and discussions for a comprehensive evaluation and understanding of our method. Detailed contents are as follows:

## A  EXPERIMENT

### A.1  MORE EXPERIMENTAL DETAILS

**Datasets.** Our experiments are conducted on a comprehensive set of benchmark datasets designed for action recognition tasks, including UCF-101 (Soomro et al., 2012), HMDB-51 (Kuehne et al., 2011), Kinetics-400 (K-400) (Carreira & Zisserman, 2017), Kinetics-600 (K-600) (Carreira et al., 2018), and Something-to-Something V2 (SSv2) (Goyal et al., 2017). The Kinetics series represents large-scale datasets with 400 and 600 distinct action classes in K-400 and K-600, respectively, where the latter expands upon the former by incorporating an additional 220 non-overlapping categories. Specifically, K-400 contains 240k training samples and 20k validation samples, while K-600 provides 410k training samples and 29k validation samples, all sourced from YouTube clips depicting human actions. Transitioning to smaller but widely adopted benchmarks, UCF-101 comprises 13,320 video samples distributed across 101 action classes, with three official train/test splits that allocate 9,537 training and 3,783 testing samples. This dataset captures diverse daily activities through short-duration videos, serving as a standard evaluation protocol in action recognition research. HMDB-51 follows a similar structure with 6,849 videos spanning 51 action categories, maintaining over 101 samples per class and three canonical splits. Notably, its content integrates clips from cinematic productions, public archives, and online platforms, offering rich contextual variations. For fine-grained interaction analysis, we evaluate on SSv2, which focuses on object-manipulation actions involving 174 semantically distinct categories. This dataset features 168,913 training samples and 24,777 testing samples, emphasizing temporally complex interactions between humans and objects that require precise spatiotemporal modeling.

**Evaluation metrics.** Building upon prior work in vision-language pre-training (Huang et al., 2024a; Yu et al., 2025a; Weng et al., 2023), we adopt two complementary evaluation frameworks to benchmark performance. For the *base-to-novel* setting, other methods train models exclusively on high-frequency action categories while testing their generalization to low-frequency counterparts. Dif-

Table 7: More ablations of Video-STAR under **cross-dataset setting**. "Split" denote evaluating across three validation splits. The best and second best performances are highlighted.

| Method | UCF-101 Split↑ | HMDB-51 Split↑ | K-600 Split↑ | Tool Library | UCF-101 Split↑ | HMDB-51 Split↑ | K-600 Split↑ |
|---|---|---|---|---|---|---|---|
| *(a)* Qwen2.5-VL-3B | 58.1±0.2 | 42.1±0.3 | 37.7±1.8 | *(g) w/o. Human* | 95.3±0.3 | 84.3±0.2 | 88.3±0.8 |
| *(b)* Video-STAR-3B | 96.7±0.3 | 86.2±0.2 | 90.5±0.7 | *(h) w/o. Pose* | 92.8±0.4 | 81.2±0.2 | 83.7±1.1 |
| *(c) SFT: w/o. Val* | 94.8±0.3 | 83.2±0.2 | 86.6±0.7 | *(i) w/o. Action* | 93.6±0.3 | 82.3±0.2 | 85.5±0.9 |
| *(d) RL: w/. Same* $w_k$ | 94.2±0.3 | 82.4±0.3 | 85.7±1.0 | *(j) w/o. Video* | 94.7±0.3 | 83.7±0.2 | 87.4±0.7 |

ferently, our method is fine-tuned only on the high-frequency action categories of HMDB-51 and evaluated in a zero-shot manner on K-400, UCF-101, and SSv2, making our evaluation more challenging. For this configuration, we apply the methodology across four benchmark datasets, including UCF-101, HMDB-51, K-400, and SSv2, systematically partitioning each into base and novel class subsets. Final results represent averages across these configurations, computed using the standard validation splits: for HMDB-51 and UCF-101, only the first of three available splits is used for novel class evaluation, while K-400 and SSv2 leverage their single validation split in full.

For the *cross-dateset setting*, we establish cross-domain evaluation protocols by strategically pairing training and testing datasets. Specifically, models trained on HMDB-51 (Kuehne et al., 2011) are evaluated on UCF-101 (Soomro et al., 2012) and K-600 (Carreira et al., 2018), while models trained on UCF-101 are tested on HMDB-51. This bidirectional transfer design enables rigorous analysis of domain adaptation capabilities. Performance characterization for HMDB-51 and UCF-101 incorporates their native three-validation-split structure, reporting both mean top-1 accuracy and standard deviation across splits. For the K-600 evaluation, we focus on 220 exclusive categories absent in Kinetics-400, adopting the standardized three-split protocol from prior work (Ni et al., 2022; Rasheed et al., 2023). Each split contains 160 novel categories, with final results representing their average performance. This multi-faceted evaluation strategy provides comprehensive insights into both within-dataset generalization and cross-dataset transfer capabilities.

**More Hyperparameter Configurations.** The proposed method was implemented on a system equipped with an Intel(R) Xeon(R) Platinum 8480+ CPU and eight NVIDIA H20 GPUs (90 GB memory each). For training, the reinforcement learning (RL) pipeline utilized the TRL framework, with supervised fine-tuning (SFT) and RL stages taking 1 hour and 20 hours, respectively.

## A.2 MORE ABLATION STUDIES

**Data Assessment.** In row *(c)* of Tab. 7, we remove the data assessment stage from our SFT training. Accuracy drops **1.9%**, **3.0%**, and **3.9%** from the full model in *(b)* on UCF-101, HMDB-51, and K-600, respectively. This indicates that filtering out inconsistent reasoning is important for maintaining data reliability and supporting tool-augmented logical coherence. Without data assessment, errors and factual inconsistencies in training data degrade cross-dataset generalization.

**Weighting** $w_k$ **in Sub-Motion Reward** $R_{sub}$. Row *(d)* evaluates the effect of assigning equal weights to all sub-motions. Compared to the full model in *(b)*, accuracy decreases **2.5%**, **3.8%**, and **4.8%** in *(d)*, showing that sub-motion-specific weighting helps prioritize semantically discriminative primitives. This confirms that our strategy of assigning dynamic importance to different sub-motions is crucial for fine-grained action reasoning.

**Tool Library**. In rows *(g)–(j)*, we analyze the contribution of each tool in our framework. Removing the human detection tool *(g)* reduces accuracy **1.4%**, **1.9%**, and **2.2%**, illustrating its role in isolating action-relevant regions and avoiding visual-semantic confusion. Without pose estimation *(h)*, performance declines **3.9%**, **5.0%**, and **6.8%**, confirming that structured kinematic representation is essential for capturing subtle motion cues. Excluding the action-retrieval tool *(i)* yields a decrease **3.1%**, **3.9%**, and **5.0%**, showing that high-level category-specific definitions help bridge label–motion gaps. Removing the video-frame description tool *(j)* lowers accuracy **2.0%**, **2.5%**, and **3.1%**, indicating that detailed temporal-segment descriptions contribute to disambiguating multi-phase actions. The full model *(b)*, which integrates the entire tool library, achieves the highest scores, proving that these tools are complementary and essential for robust performance.

Table 8: Ablation on the teacher model for Chain-of-Thought (CoT) data generation.

| CoT Generation Model | UCF Acc↑ | Delta |
|---|---|---|
| Qwen2.5-VL-72B (Ours) | 96.7% | - |
| InternVL-2.5-78B | 97.1% | +0.4% |
| Gemini-1.5-Pro | 97.5% | +0.8% |

**Ablation on CoT Data Generation.** To confirm that our framework's effectiveness is not limited to a specific model family or a result of intra-family knowledge distillation, we experimented with generating the CoT data using different powerful MLLMs. As shown in Table 8, when using InternVL-2.5-78B or Gemini-1.5-Pro as the "teacher" model to generate training data, the performance of Video-STAR further improves to 97.1% and 97.5%, respectively. This not only demonstrates the robustness and generalizability of our approach but also highlights that our framework's performance can scale with the reasoning capability of the teacher model used for data generation.

Table 9: Comparison with state-of-the-art (SOTA) frontier models.

| Method | K-400↑ | HMDB-51↑ | UCF-101↑ |
|---|---|---|---|
| Qwen2.5-VL-7B | 86.3% | 45.6% | 83.7% |
| Qwen3-VL-8B | 89.5% | 68.5% | 87.1% |
| Gemini-1.5-Pro | 92.8% | 73.6% | 93.3% |
| **Video-STAR-7B** | 96.7% | 92.1% | 99.7% |

**Comparison with Frontier SOTA Models.** To position our work against the latest advancements, we benchmarked Video-STAR against powerful, general-purpose frontier models like Qwen3-VL and Gemini-1.5-Pro. The results in Table 9 are decisive. Even a highly capable model like Gemini-1.5-Pro struggles with the fine-grained discrimination required by HMDB-51, scoring only 73.6%. In stark contrast, our Video-STAR-7B achieves 92.1%, a massive 18.5% absolute improvement. This large performance delta across all benchmarks proves that for specialized tasks like OVAR, a sophisticated, task-specific reasoning framework is far more critical and effective than simply relying on a larger, general-purpose model.

## A.3 MORE CASE STUDIES

As shown in Fig. 5 and Fig. 6, we present more case studies between Qwen2.5-VL-3B and our Video-STAR-3B model on the HMDB-51 dataset. Our method can effectively leverage domain-specific tools combined with sub-action decomposition to achieve fine-grained action recognition.

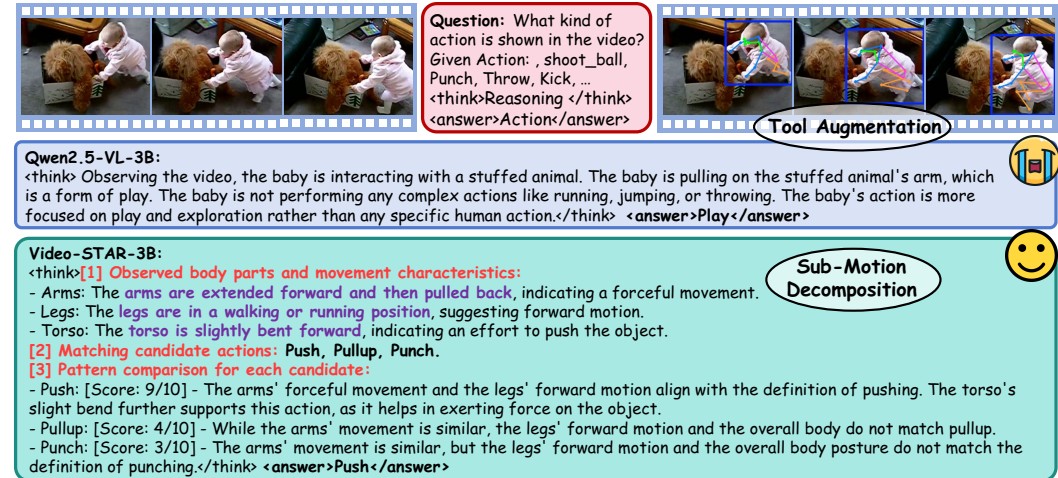

Figure 5: **More Case Study between Qwen2.5-VL-3B and our Video-STAR-3B.** Qwen2.5-VL-3B misclassifies "push" as "play", while our Video-STAR-3B accurately identifies the correct action.

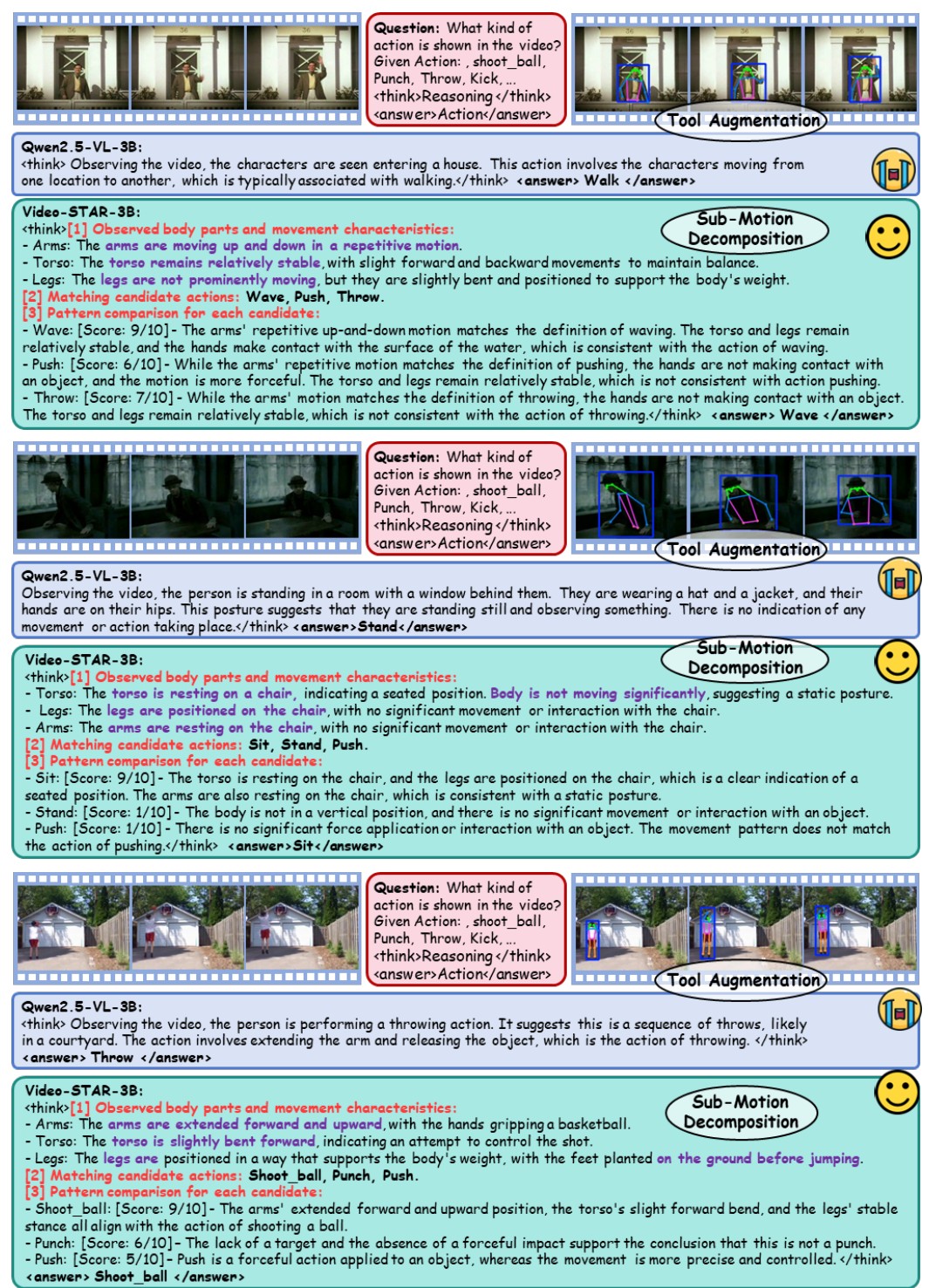

Figure 6: **More Case Study between Qwen2.5-VL-3B and our Video-STAR-3B.** Our Video-STAR-3B can accurately identify the correct action.

## B RELATED WORK

### B.1 OPEN-VOCABULARY ACTION RECOGNITION.

Building on the robust feature extraction capabilities of CLIP's pre-training, researchers have devised numerous video recognition methods leveraging its cross-modal alignment strengths. Initial approaches primarily focused on adapting CLIP via full fine-tuning or parameter-efficient strategies. For instance, ActionCLIP (Wang et al., 2021) pioneered the adaptation of CLIP for video recognition by fine-tuning and augmenting the entire model to capture motion. Similarly, STAN (Liu et al., 2023a) introduced auxiliary networks to extract temporal features from CLIP's frozen representations, while ViFi-CLIP (Rasheed et al., 2023) simultaneously fine-tuned both visual and text encoders to achieve better performance. These methods, though effective in closed-set scenarios, often overfit to static cues due to extensive parameter updates.

There are two primary paradigms for parameter-efficient paradigms: architecture adaptation and prompt engineering. Adaptation-based methods like Adaptformer (Chen et al., 2022), X-CLIP (Ni et al., 2022), and ST-Adapter (Pan et al., 2022) employ compact neural modules in CLIP to distill temporal knowledge from static image representations. Prompt engineering approaches like AIM (Yang et al., 2023) and DualTrans (Park et al., 2023), which inject learnable prefix tokens into CLIP's text encoder to bias the model towards action-specific semantics, while VPT (Ju et al., 2022) and Vita-CLIP (Wasim et al., 2023) explore both prefix and suffix prompting strategies to align video features. Despite these, their performance is limited by adapters' limited capacity, prompt-based modulations, and architectural rigidity, which hinders cross-architecture generalization.

Recent efforts explicitly targeted open-vocabulary generalization. Open-VCLIP (Weng et al., 2023) introduced weight interpolation to regularize fine-tuning, but this approach restricted compatibility to architectures with identical weight dimensions. FROSTER (Huang et al., 2024b) further advanced this by leveraging residual feature distillation. Open-MeDe (Yu et al., 2025a) adopts a meta-learning framework that explicitly mitigates static bias via cross-batch meta-optimization, yet their dependency on CLIP's static generalization hindered robustness to motion-centric tasks. Differently, our work diverges from these paradigms by proposing a framework that integrates sub-motion decomposition with tool-augmented RL, thereby enabling robust open-vocabulary inference.

### B.2 MULTIMODAL LLMS REASONING.

Recent advances in large language models (LLMs) have demonstrated that RL-based post-training can significantly enhance reasoning capabilities, as exemplified by OpenAI-o1 (Jaech et al., 2024) and DeepSeek-R1 (Guo et al., 2025). Several works have extended these paradigms to multimodal language models (MLLMs) for tasks such as mathematical and scientific image VQA (Peng et al., 2025; Huang et al., 2025), image segmentation and grounding (Liu et al., 2025a; Bai et al., 2025b; Shen et al., 2025; Liu et al., 2025b), video spatial or temporal grounding (Wang et al., 2025d; Ge et al., 2025; Park et al., 2025; Li et al., 2025c) and video understanding VQA (Feng et al., 2025; Wang et al., 2025b; Li et al., 2025b; Cheng et al., 2025). Despite these, existing methods remain limited in cross-modal interaction and hallucination, especially for long video scenarios (Chen et al., 2025). To address this, we propose multimodal CoT reasoning through tool-augmented RL, which explicitly reduces hallucination through structured perception.

### B.3 TOOL-AUGMENTED AGENTIC SYSTEM.

Recent advancements in LLMs have demonstrated that integrating external tools can significantly enhance multimodal reasoning capabilities. Early works like FAST (Sun et al., 2025) and MVoT (Li et al., 2025a) established foundational approaches by incorporating visual evidence into reasoning processes, forming multimodal CoT for image tasks. Concurrently, LLaVa-Plus (Liu et al., 2023b) pioneered training strategies for tool use in visual reasoning, while Visual Program Distillation (Hu et al., 2024) leveraged program-derived CoT data to transfer tool-use skills. Visual CoT (Shao et al., 2024) further enriched this domain by creating synthetic datasets to boost reasoning. CogCoM (Qi et al., 2024) introduced six manipulation strategies through synthetic chain-of-manipulation (CoM) data, and TACO (Ma et al., 2024) provided 273K multimodal reasoning traces from 15 visual tools. Recent methods like Simple (Wang et al., 2025c), THYME (Zhang et al., 2025b), and PyVision (Zhao et al., 2025b) extended tool use with RL, thus enhancing general

reasoning capacity. Visual search capabilities were advanced by DeepEyes (Zheng et al., 2025), VGR (Wang et al., 2025a), and OpenThinkImg (Su et al., 2025b), which integrated tools such as image zoom-in and sketching to improve visual analysis. However, partial methods often rely on static tool pipelines, limiting their ability to disentangle complex actions in open-vocabulary settings. Our framework addresses this by explicitly modeling contextual sub-motions and dynamically balancing tool usage efficiency with hierarchical motion relevance.

## C  DISCUSSION

**Enforcing Visual Grounding via Tool-Augmented Visual CoT.** A core objective of our framework is to mitigate the "text-centric reasoning" prevalent in many MLLMs. We achieve this by enforcing a *Visual Chain-of-Thought* (Visual CoT), a process fundamentally driven by our tool-use mechanism. Instead of allowing the model to generate abstract, text-based reasoning steps, we compel it to build its logic from concrete visual evidence extracted by external tools. Specifically, tools like pose estimation and human detection extract structured, non-textual facts (e.g., skeletal keypoints, bounding boxes) directly from the video frames. These geometric facts become the foundational "thoughts" in the model's reasoning process. For instance, rather than a vague thought like "the person is moving their arm," the model is guided to construct a thought grounded in visual evidence, such as: "The wrist keypoint [x1, y1] rises from a low position to above the shoulder keypoint [x2, y2], then oscillates horizontally." By designing a reward function that prioritizes reasoning chains demonstrably built upon these tool-generated visual facts, we effectively force the model's entire inference process to be anchored to the visual world, preventing it from drifting into pure text-based speculation and reducing cross-modal hallucinations.

**Discussion on Computational Cost and Future Work.** While Video-STAR demonstrates a highly favorable trade-off between performance and cost, we acknowledge the computational overhead introduced by tool invocation, which accounts for a significant portion of the inference latency (as shown in Table 5). This latency is primarily due to executing external models and making API calls. However, we view this as a promising direction for future optimization rather than a fundamental limitation. For practical deployment, inference time can be significantly accelerated through standard engineering techniques. Model distillation (e.g., from a 3B to a 1B model), quantization (e.g., to 8-bit or 4-bit precision), and parallelization strategies (e.g., tensor and pipeline parallelism) could drastically reduce both memory footprint and latency. Exploring these optimizations to create a lightweight yet powerful version of Video-STAR presents an exciting avenue for future research.

## D  TOOL LIBRARY DETAILS

This section provides a detailed specification of tools used by our Video-STAR framework, including human detection tools, pose estimation tools, and online retrieval-augmentation tools.

### D.1  HUMAN DETECTION TOOLS

Our framework employs the Ultralytics YOLO 11 (Weng et al., 2023) detector as a human detection model. We chose this model for its optimization for real-time performance and accuracy. Its primary function is to locate people in video frames. The model employs an anchor-free detection architecture combined with spatial attention mechanisms. This design enables it to handle common challenges such as occlusions, varying human scales, and crowded scenarios. The tool's ability to ensure precise localization of human bounding boxes, even in cluttered environments, is critical for our framework. It allows our model to isolate action-relevant regions from background distractions, which is a key step in reducing visual-semantic ambiguity and preventing model hallucination during the reasoning process.

### D.2  POSE ESTIMATION TOOLS

For pose estimation, we utilize the skeletonization capability of Ultralytics YOLO 11, which translates a detected person into a structured kinematic representation. The model outputs a skeleton

with 17 keypoints that follow the COCO format, including joints for the elbows, wrists, and knees. It achieves this with sub-pixel precision, which is essential for detailed motion analysis. The model infers these joint positions using its knowledge of hierarchical spatial relationships between body parts. For instance, it understands the natural correlation between hip and knee angles during loco-motion. This underlying knowledge allows it to generate coherent skeletons even when body parts are in fast motion or partially occluded. This fine-grained representation enables our system to capture subtle motion patterns. For example, we can analyze the precise wrist rotation in a "handshake" or the exact knee flexion during a "jump".

### D.3 ONLINE RETRIEVAL-AUGMENTATION TOOLS

We use the Qwen API as our online retrieval-augmentation tool. This tool provides our framework with essential contextual information. It has two primary functions, ❶ The first function is to resolve the ambiguity of high-level action definitions. We use this tool to get a detailed, category-specific explanation for a given action label. It helps redefine action words through their transitional phases (*e.g.*, "stand" is described as "transit from sit or lay to stand"). This process provides our model with a deeper semantic understanding, closing the gap between a simple label and its complex physical meaning. ❷ The second function is to describe the motion within specific key frames, which helps the model understand complex, multi-phase movements. After our framework extracts temporally salient frames (*e.g.*, frames with maximum joint displacement), we send these frames to the tool. The tool then generates a concise textual description of the action happening at that moment. This description emphasizes both spatial configurations, like "rapid arm extension," and temporal dynamics, like "followed by torso rotation". This maps descriptive text precisely to informative video segments, allowing the model to disambiguate sequential actions by understanding each component.

## E PROMPTS

During the supervised fine-tuning (SFT) stage of Video-STAR, we designed a structured input prompt to generate high-quality chain-of-thought (CoT) data. The prompt template is as follows:

**First Round Prompts:**

```
FIRST_TURN_TEMPLATE =
("""
I will show you a video of human action. Your task is to determine which
    visual analysis tools would help you better identify the action. You
    have three independent tools available:

1. **Pose Estimation Tool**: Adds skeleton keypoints and connections to
    show body pose and joint movements
2. **Person Detection Tool**: Adds bounding boxes around detected persons
     to highlight human subjects
3. **Noun Explanation Tool**: Provides detailed explanations of action
    types to help with classification
4. **Video Description Tool**: Provides description of input video to
    help better understand video content

Please analyze the video using step-by-step reasoning and decide for each
     tool independently:

Analysis Requirements:
1. **For Pose Estimation**: Assess whether body joint movements and pose
    details are crucial for identifying this action
2. **For Person Detection**: Evaluate whether clearly identifying and
    localizing the person(s) would help with action recognition
3. **For Noun Explanation**: Consider whether you need detailed
    explanations of action categories to make accurate classification
4. **For Video Description**: Judge whether you require a detailed
    description concerning video to help identifying action

Output Format:
<think>step-by-step reasoning process:
```

```
[1] Video content analysis: [describe what you observe in the video]
[2] Pose estimation evaluation: [assess whether joint/skeleton info would
     help]
[3] Person detection evaluation: [assess whether person localization
    would help]
[4] Noun explanation evaluation: [assess whether action category details
    would help]
[5] Final tool selection reasoning: [explain your decisions for each tool
    ]
</think>
<action>
<human>yes or no</human> <pose>yes or no</pose>
<action>yes or no</action> <video>yes or no</video>
</action>
""")
```

**Second Round Prompts:**

```
SECOND_TURN_TEMPLATE =
("""
I will provide you with a video and ask you to identify the human action
    shown.
There is also an annotated version of this video available.
The annotated video may contains the same content but with additional
    annotations:
- Blue bounding boxes around detected persons
- Pose estimation points and skeleton lines showing body keypoints

Your task is to follow this three-stage reasoning process:

Input Format:
Original Question: What kind of human action is shown in the video?
A Description of this Video: {VIDEO_DESCRIPTION}
All action types and their explanations are shown as follows: {
    ALLOWED_ANSWERS}.

Analysis Requirements:
**Stage 1: Sub-Motion Decomposition**
1. Identify all body parts showing significant motion in the video (e.g.,
    arms, legs, torso)
3. For each identified body part:
   - Note its movement direction (up/down, rotational, etc.)
   - Record contact points with objects (if any)
3. List them by importance from top to bottom

**Stage 2: Action Candidate Selection**
1. Extract the key movement patterns from its description in given action
    types
2. Compare with the video's observed:
   - Temporal sequence of movements (which body part moves first)
   - Interaction patterns between body parts
   - Force application points (e.g., hand gripping vs. pulling)
3. Generate 2-3 candidate actions that best match the observed body part
    movements

**Stage 3: Matching Scoring**
1. Match these observations to the action definitions in given action
    types
2. Score each candidate based on: (Maximum Score: 10)
   - Body part involvement precision
   - Movement pattern similarity
   - Object interaction consistency

Output Format:
<think>step-by-step reasoning process:
```

```
[1] Observed body parts and movement characteristics:
   - [Body Part 1]: [Direction/Contact Description]
   - [Body Part 2]: [Direction/Contact Description]
   ...
[2] Matching candidate actions:
   - [Candidate 1]: Matches [Body Part A] [Movement Type]
   - [Candidate 2]: Matches [Body Part B] [Movement Type]
   ...
[3] Pattern comparison for each candidate:
   - [Candidate 1]: [Score] - [Matching Details]
   - [Candidate 2]: [Score] - [Matching Details]
</think>
<answer>action-type</answer>
""")
```

This structured prompt ensures our dataset contains diverse and causally plausible reasoning processes, which are critical for cultivating the model's foundational perception and planning capabilities before RL fine-tuning.

## F    LLM CLARIFICATION

We clarify the role of Large Language Models (LLMs) in the development of this manuscript. During the drafting process, LLMs were utilized primarily for two purposes: translating initial technical content from Chinese to English and refining the language of the final draft. This included correcting grammatical inconsistencies, optimizing sentence structure, and improving the clarity and coherence of the narrative. It is essential to emphasize that all conceptual contributions, including the formulation of open-vocabulary action recognition challenges, the design of the Video-STAR framework (e.g., contextual sub-motion decomposition and hierarchical reward mechanisms), the integration of domain-specific tools (e.g., YOLO 11 for pose estimation and Qwen API for explanations), and the experimental design across HMDB-51, UCF-101, and Kinetics-400/600 datasets, are the original work of the human authors. The LLM was employed strictly as a linguistic tool to enhance the presentation of ideas, with no involvement in hypothesis generation, algorithmic innovation, or data analysis. The core scientific claims, methodological innovations, and empirical validation presented in this paper is entirely attributable to the research team.

