# OpenReview forum: "Video-STAR: Reinforcing Open-Vocabulary Action Recognition with Tools"
_ICLR.cc/2026/Conference — ICLR 2026 Poster_

### Official Review · Reviewer_zTP9 · 2025-10-29

**Soundness:** 3
**Presentation:** 3
**Contribution:** 3
**Rating:** 6
**Confidence:** 4

**Summary:**

The paper proposes Video-STAR, a framework for open-vocabulary action recognition that integrates contextual sub-motion decomposition with tool-augmented reinforcement learning on top of MLLMs. Actions are no longer treated as atomic labels; instead, they are decomposed into discriminative motion primitives, while external tools are invoked to reduce cross-modal hallucinations and enable category-specific reasoning. A hierarchical reward is designed to jointly optimize tool-usage efficiency, structural coherence, and sub-motion relevance. Extensive experiments demonstrate substantial gains over CLIP-based baselines and vanilla MLLMs across multiple settings.

**Strengths:**

The combination of sub-motion decomposition and tool-augmented RL constitutes a meaningful departure from static cross-modal alignment pipelines commonly used in OVAR.

The method delivers consistent and large improvements across diverse benchmarks and evaluation settings.

**Weaknesses:**

1. While the integration is well-designed, the core building blocks (tool-augmented CoT, RL-based post-training, and sub-action decomposition) are all known paradigms; the novelty is primarily at the system-level composition rather than at the level of a fundamentally new principle.

2. The approach depends on specific external tools, yet the paper does not analyze robustness to tool inaccuracies or the portability of the method under alternative tool choices.

3. The computational overhead of repeated tool invocation and multi-round RL inference is not reported nor compared against CLIP-based or purely SFT-based OVAR pipelines. Furthermore, the training cost of GRPO fine-tuning is also not quantified or compared with prior methods.

4. Although standard CLIP-based OVAR methods are included, the paper does not benchmark against recent LLM-augmented CLIP paradigms that explicitly incorporate generative priors for action understanding, such as [1–3].

5. The evaluation is conducted primarily against mid-scale or earlier-generation MLLMs (e.g., Qwen2.5-VL), without comparison to state-of-the-art frontier models (e.g., Qwen3-VL, GPT-5, Claude, Gemini), many of which already demonstrate strong video reasoning capabilities.

[1] Building a Multi-modal Spatiotemporal Expert for Zero-shot Action Recognition with CLIP

[2] Generating Action-Conditioned Prompts for Open-Vocabulary Video Action Recognition

[3] VTD-CLIP: Video-to-Text Discretization via Prompting CLIP

**Questions:**

Please see the weakness.

---

> ### Author Response · Authors · 2025-11-22
> **Rebuttal by Authors**
>
> Sincerely thanks for your positive review. We address your concerns below.
>
> > **Q1**: The core building blocks are all known paradigms; the novelty is primarily at the system-level composition rather than at the level of a fundamentally new principle.
>
> **A1**: Our framework's novelty lies not in merely listing these features, but in their synergistic design where reinforcement learning trains an **agentic** system. This agent learns to dynamically invoke tools to generate the structured visual evidence needed for a robust sub-motion decomposition. Then the reasoning process is optimized by a hierarchical reward, ensuring the effective differentiation between semantically similar, unseen actions. As the **first agentic reasoning framework in OVAR field**, our innovation has already been highlighted and appreciated by `reviewer mh4G` and `reviewer vdeG`.
>
> **Table 1: Base-to-Novel Comparison**
>
> |Method|K-400↑|HMDB-51↑|UCF-101↑|
> |---|---|---|---|
> |Open-MeDe (ICCV 2025) [1]|69.9%|63.9%|85.9%|
> |FROSTER (ICLR 2024) [2]|70.4%|65.1%|87.0%|
> |**Video-STAR-7B (Ours)**|**96.7%**|**92.1%**|**99.7%**|
>
> Moreover, as shown in `Table 1`, our method pushes the performance to **near-perfect scores** on established datasets like K-400 (**96.7%**), HMDB-51 (**92.1%**) and UCF-101 (**99.7%**), achieving absolute improvements of **+26.3%**, **+27.0%** and **+12.7%** respectively over FROSTER [2]. Such a significant improvement demonstrates that our novel innovation is not incremental, but a breakthrough in solving these benchmark challenges.
>
> > **Q2**: Analyze robustness to tool inaccuracies or the portability of the method under alternative tool choices.
>
> **A2**: Thanks for your question. Our method was intentionally designed with a **decoupled, tool-agnostic** system to ensure both portability and flexiblility of selected tools. Each tool functions as a modular "information provider", allowing our **agent system** to focus on a general policy of "reason with evidence" rather than learning dependencies on a specific tool.
>
> To address your concerns, we validated this by swapping YOLO 11 with OpenPose for pose estimation and Qwen API with Gemini-1.5-Pro for action explanation​, as shown in `Table 2` below.
>
> **Table 2: Ablations on Tool Selection**
>
> |Tool Stack (3B)|UCF Acc↑|Latency (per sample)|
> |---|---|---|
> |YOLO 11 + Qwen (Ours)|96.9%|3.18s|
> |YOLO 11 → OpenPose|96.1%|4.61s|
> |Qwen → ​Gemini-1.5-Pro​|97.4%|3.47s|
>
> As can be seen, the slight variations in accuracy and latency (e.g., OpenPose is **slower** but still effective; Gemini-1.5-Pro is better but more **expensive**) demonstrate our method's modularity. The consistently high performance across different tool combinations confirms that the primary driver of our results is the underlying agentic logic​, not a fragile dependency on a single tool. This decisively proves the effectiveness of our ​agentic reasoning framework​, which can flexibly orchestrate a variety of **alternative** tools.
>
> The results are presented in Table 6 (`Line 502`) and analyzed in the **Robustness to Tool Selection** part (`Sec 4.2`, `Line 508`) of our revised paper.

---

> ### Author Response · Authors · 2025-11-22
> **Rebuttal by Authors**
>
> > **Q3**: The computational overhead of repeated tool invocation and multi-round RL inference is not reported. The training cost of GRPO is also not quantified or compared with prior methods.
>
> **A3**: Thanks for your advice. We would like to clarify a potential misunderstanding about "repeated tool invocation and multi-round inference" Our agentic system employs a two-round inference process per sample, not multiple rounds of calls. In the first round, the model analyzes the input and autonomously decides which tools to invoke. In the second round, it integrates the outputs from all selected tools to perform the final reasoning. This is a one-time cost for tool usage during inference.
>
> Following your advice, we have evaluated our model's computational overhead and compared it with its base model, as shown in `Table 3` below:
>
> **Table 3: Computational Overhead**
>
> |Method|Memory↓|Latency↓|UCF Acc↑|GFLOPs|BSZ|
> |---|---|---|---|---|---|
> |Qwen-2.5VL-3B|60.03 GB|0.79s + 0s|**58.1%**|18,296|50|
> |Video-STAR-3B|60.89 GB|1.75s + 1.43s|**96.7%**|25,773|50|
>
> Note that experiments were conducted on a single NVIDIA H20 GPU, with both GFLOPs and Latency measuring per-sample results. Latency is split into inference time and tool invocation time. As can be seen, the increase of latency is primarily due to tool invocation, which accounts for 1.43s of the total 3.18s latency. This time is primarily spent on executing external models like YOLO for pose/human detection and making API calls for descriptions. Despite this, the boosting accuracy from **58.1%** to **96.7** effectively highlights a highly favorable **trade-off** between computational cost and performance.
>
> The above results are presented in Table 5 (`Line 490`) and analyzed in the **Computational Overhead** part (`Sec 4.2`, `Line 495`) of our revised paper.
>
> Regarding the training cost of GRPO, the training was conducted on eight NVIDIA H20 GPUs (90 GB memory each) using a 5k sample dataset with a batch size of 8. The process took 20 hours for 600 iterations (1 epoch), with 4 rollouts per sample and a learning rate of 5e-7. These details can be found in `Sec 4` (`Line 368`).
>
> > **Q4**: The paper does not benchmark against recent LLM-augmented CLIP paradigms that explicitly incorporate generative priors for action understanding, such as [1–3].
>
> **A4**: Thank your for your advice. We have benchmarked our results against them, shown as below:
>
> **Table 4: Base-to-Novel Setting Comparison**
>
>
> |Method|K-400↑|HMDB-51↑|UCF-101↑|SSv2↑|
> |---|---|---|---|---|
> |VTD-CLIP (ArXiv 2025) [3]|70.1%|70.0%|83.2%|15.4%|
> |AP-CLIP (ACMMM 2024) [4]|70.0%|67.0%|88.5%|14.3%|
> |Video-STAR-3B|86.2%|91.9%|97.9%|12.3%|
> |**Video-STAR-7B**|**96.7%**|**92.1%**|**99.7%**|**15.5%**|
>
> **Table 5: Cross-Dataset Setting Comparison**
>
> |Method|UCF-101↑|HMDB-51↑|K-600↑|
> |---|---|---|---|
> |STDD (AAAI 2025) [5]|85.2±1.2%|55.9±0.2%|75.1±0.7%|
> |Video-STAR-3B|96.7±0.3%|86.2±0.2%|90.5±0.7%|
> |**Video-STAR-7B**|**99.4±0.2%**|**92.5±0.1%**|**98.2±0.1%**|
>
> As can be seen, in the base-to-novel setting (`Table 4`), our Video-STAR-7B surpasses VTD-CLIP by an astronomical **+26.6%** on K-400 and AP-CLIP by **+25.1%** on HMDB-51. In the cross-dataset setting (`Table 5`), our 7B model outperforms STDD by **+14.2%** on UCF-101 and **+36.6%** on HMDB-51. Even our smaller 3B model outperforms all these methods across the board.
>
> We have updated these SOTA methods for comparison in both Table 1 (**VTD-CLIP** in `Line 335`, **AP-CLIP** in `Line 336`) and Table 2 (**STDD** in `Line 390`) of the revised version.

---

> > ### Author Response · Authors · 2025-11-27
> > **Rebuttal by Authors**
> >
> > > **Q5**: The evaluation lacks comparison to SOTA frontier models (e.g., Qwen3-VL, GPT-5, Gemini).
> >
> > **A5**: Thank you for your feedback. At the time of our initial submission, models like Qwen3-VL were not yet released. We have now evaluated the zero-shot performance of the newly released Qwen3-VL and the powerful Gemini-1.5-Pro​, as shown in `Table 6` below:
> >
> > **Table 6: Comparison with SOTA Frontier Models**
> >
> > |Method|K-400↑|HMDB-51↑|UCF-101↑|
> > |---|---|---|---|
> > |Qwen2.5-VL-7B|86.3%|45.6%|83.7%|
> > |Qwen3-VL-8B|89.5%|68.5%|87.1%|
> > |Gemini-1.5-Pro​|92.8%|73.6%|93.3%|
> > |**Video-STAR-7B**|**96.7%**|**92.1%**|**99.7%**|
> >
> > As can be seen, even a powerful frontier model like Gemini-1.5-Pro struggles with the fine-grained discrimination in HMDB-51 dataset, scoring 73.6%. In contrast, our Video-STAR framework achieves an exceptional 92.1%, a massive **18.5%** absolute improvement. This trend holds across all benchmarks, including a **6.4%** lead on UCF-10. This large performance delta decisively proves that for specialized tasks like OVAR, a sophisticated reasoning is far more critical than simply relying on a general-purpose frontier model.
> >
> > The results are presented in Table 7 (`Line 516`) and analyzed in the **Comparison with Frontier SOTA Models** part (`Sec 4.2`, `Line 522`) of our revised paper.
> >
> > **Reference**
> >
> > [1]. Yu, Y., Cao, C., Zhang, Y., & Zhang, Y. (2025). Learning to Generalize without Bias for Open-Vocabulary Action Recognition. arXiv preprint arXiv:2502.20158.
> >
> > [2]. Huang, X., Zhou, H., Yao, K., & Han, K. (2024). FROSTER: Frozen CLIP is a Strong Teacher for Open-vocabulary Action Recognition. In The Twelfth International Conference on Learning Representations (ICLR).
> >
> > [3]. Zhu W, Wang Y, Li H, et al. VTD-CLIP: Video-to-Text Discretization via Prompting CLIP[J]. arXiv preprint arXiv:2503.18407, 2025.
> >
> > [4]. Jia C, Luo M, Chang X, et al. Generating action-conditioned prompts for open-vocabulary video action recognition[C]//Proceedings of the 32nd ACM International Conference on Multimedia. 2024: 4640-4649.
> >
> > [5]. Yu Y, Cao C, Zhang Y, et al. Building a multi-modal spatiotemporal expert for zero-shot action recognition with clip[C]//Proceedings of the AAAI Conference on Artificial Intelligence. 2025, 39(9): 9689-9697.

---

### Official Review · Reviewer_vdeG · 2025-10-31

**Soundness:** 3
**Presentation:** 3
**Contribution:** 3
**Rating:** 8
**Confidence:** 4

**Summary:**

This paper proposes, implements and investigates a novel strategy and framework, i.e., Tools Augmented CoT reasoning for zero-shot fine-grained human action recognition. The framework employs VFMs, VLMs and Visual RAG to extract sub-concept vision representations such as human, pose, and video explanation, and generated a combined prompt on proposed format for CoT to MLLM for final stage prediction. The core innovations are the format definition and implementation of sub-action decomposition, candidate selection, and matching scoring for CoT and reinforcement learning. Concrete evaluations are performed on five formal benchmarks and the results shown significant improvements over the SOTA.

**Strengths:**

A novel Tool Augmented CoT framework and implementation approach for open-vocabulary action recognition (OVAR), concrete evaluations of 5 benchmarks and new SOTA performance.

**Weaknesses:**

There are still some uncertain issues. (1) the experiments on only one base MLLM model, Qwen2.5-VL, are reported, where the training prompts of reasoning chain for CoT are generated by Qwen2.5-VL-72B, and then fine-tune the small-size model Qwen2.5-VL-3B and Qwen2.5-VL-7B for experiments, is teacher-student knowledge distillation on the proposed reasoning chain format able to achieve similar effectiveness? May be better to add more results on other leading frontier MLLMs such as InternVL2.5, Gemini-2.5-Pro, Llama, etc. (2) is it applicable to professional actions such as FineGym, Diving? Where expertise sub-action concepts might not well be learned for AGI models. (3) On lines 356-360, are the protocols defined for previous benchmarks? Please cited them. If novel classes Y_N are completed unknown in MLLM, how to generate the novel nouns of the new classes? So that the base VLM and MLLM have been trained on related concepts, and may be not strictly zero-shot performance. (4) As the training sub-concepts and reasoning chains are generated by VFM, VLM, and MLLM, are there hallucinated chains which lead to final correct answers on ground truth? Maybe the discussion on the Visual Grounded CoT is helpful.

**Questions:**

See Weaknesses.

---

> ### Author Response · Authors · 2025-11-22
> **Rebuttal by Authors**
>
> Thank you for your positive and encouraging review. Your insightful summary demonstrates a deep understanding of our work's core contributions. We address your concerns below.
>
> > **Q1**: Using other leading frontier MLLMs like InternVL2.5 and Gemini-1.5-Pro for generating CoT.
>
> **A1**: Thank your for your suggestion. Following your advice, we conducted new experiments where the Chain-of-Thought (CoT) data was generated by two different leading MLLMs: **InternVL-2.5-78B** and ​**Gemini-1.5-Pro**​, instead of our original Qwen2.5-VL-72B. We then used this new data to fine-tune our Video-STAR-3B model. The results are as shown in `Table 1` below.
>
> **Table 1: Ablations on CoT Generation**
>
> |CoT Generation|UCF Acc↑|Delta|
> |---|---|---|
> |Qwen2.5-VL-72B (Ours)|96.7%|-|
> |InternVL-2.5-78B|97.1%|+0.4%|
> |Gemini-1.5-Pro|97.5%|+0.8%|
>
> The results confirm that the framework's effectiveness is not limited to a specific model family. Using InternVL and Gemini as "teacher" models further improved performance to 97.1% (**+0.4%**) and 97.5% (**+0.8%**), respectively. This demonstrates that the performance gains stem from our proposed reasoning framework itself, rather than intra-family knowledge distillation. Moreover, it highlights that our method's performance scales with the reasoning capability of the teacher model used for data generation, proving the robustness and generalizability of our approach.
>
> The results are presented in Table 9 (`Line 920`) and analyzed in the **Ablation on CoT Data Generation** part (`Appendix A.2`, `Line 925`) of our revised paper.
>
> > **Q2**: Is it applicable to professional actions such as FineGym, Diving? Where expertise sub-action concepts might not well be learned for AGI models.
>
> **A2**: Yes, our framwork is well applicable to professional actions. The training's primary role is to incentivize the model's capabilities on how to invoke tools and **decompose actions**. Therefore, the model learns the ​skill of analyzing these universal primitives​, not memorizing their seen combinations in training data. Moreover, the unique algorithm characteristics of RL makes our approach highly effective for **generalization**, as evidenced by by recent research [1] ("SFT Memorizes, RL Generalizes"). Therefore, our framework excels by decomposing both the visual cues and the candidate action definitions into the **same primitive space** for a fine-grained comparison. In contrast, generic AGI models might struggle to differentiate these concepts, thus leading to misidentification.
>
> Furthermore, from a data perspective, our framework has already demonstrated strong capabilities on the very domains mentioned. Specifically, the **UCF-101** test set includes "**Floor Gymnastics**" (class #35), and the **HMDB-51** test set includes "**Diving**" (class #7). Our Video-STAR-3B model achieves remarkable novel-set accuracy of **98.9%** on UCF-101 and **91.9%** on HMDB-51 (Table 1). Upon further inspection, we found that the recognition accuracy for these two specific classes is both **​100%**.
>
> This demonstrates that our framework’s ability to decompose complex motions is robust enough to handle expert-level actions, supporting its potential for broader application in specialized fields.
>
> > **Q3**: On lines 356-360, are the protocols defined for previous benchmarks? Please cited them. If novel classes Y_N are completed unknown in MLLM, how to **generate** the novel nouns of the new classes?
>
> **A3**: Thanks for your questions. The protocols of both base-to-novel and cross-dataset setting are standard, which can be found in previous papers including **FROSTER (ICLR 2024)** [2] and ​**Open-MeDe (ICCV 2025 Highlight)** [3]​. This ensures that our results are directly comparable to the latest advances in the field. We have updated our paper with these citations.
>
> For zero-shot nature, open-vocabulary action recognition is fundamentally a ​**discriminative task**, not a **generative one​**. Its goal is ​**not to generate** answers from scratch for completely unknown concepts​. Instead, during inference, the model is given a list of candidate actions (unseen during training) and then decide which one **best matches** the video. This means the model's challenge is not to "generate" a novel action, but to effectively **differentiate** between the textual definitions of candidates (e.g., "diving" vs. "high jump") based on the visual sub-motions it observes. Our framework excels by decomposing both the visual cues and the candidate action definitions into the same primitive space for a fine-grained comparison.

---

> ### Author Response · Authors · 2025-11-22
> **Rebuttal by Authors**
>
> > **Q4**: Are there hallucinated chains which lead to final correct answers on ground truth?
>
> **A4**: Thanks for your question. We anticipated the potential for "hallucinated chains" and implemented a rigorous filtering process. Specifically, after generating the CoT data, we employed the closed source **Qwen-VL-Max** as an expert judge to score and reject any flawed chains, ensuring coherence and factual accuracy. This automated review is complemented by **human verification**, where we perform manual checks on 10% of these filtered chains to ensure ultimate reliability. This dual-validation approach guarantees the high fidelity of the approximately 5,000 chains that constitute our final dataset.
>
> This process has been detailed in `Sec 3.2` (`Line 247-Line 257`) of the revised version.
>
> **Reference**
>
> [1]. Chu T, Zhai Y, Yang J, et al. SFT Memorizes, RL Generalizes: A Comparative Study of Foundation Model Post-training[C] //Forty-second International Conference on Machine Learning.
>
> [2]. Huang, X., Zhou, H., Yao, K., & Han, K. (2024). FROSTER: Frozen CLIP is a Strong Teacher for Open-vocabulary Action Recognition. In The Twelfth International Conference on Learning Representations (ICLR).
>
> [3]. Yu, Y., Cao, C., Zhang, Y., & Zhang, Y. (2025). Learning to Generalize without Bias for Open-Vocabulary Action Recognition. arXiv preprint arXiv:2502.20158.

---

### Official Review · Reviewer_Hc9X · 2025-11-01

**Soundness:** 3
**Presentation:** 3
**Contribution:** 2
**Rating:** 4
**Confidence:** 3

**Summary:**

The paper discusses this idea: don’t let the model guess in one shot. Make it first break the action into sub-motions (arms, torso, legs, contact), then match those to candidate actions, and finally score them — and let the model call tools (YOLO human detection, pose estimation, Qwen-based action/video explanation) when it thinks vision cues are not enough.

**Strengths:**

1. The paper names two real OVAR pain points, cross-modal hallucination and similar-action confusion, and every design choice (tools, sub-motions, hierarchical reward) points back to those, making a coherent motivation.
2. Treating “shoot ball” as “bend → jump → arm extend → release” matches how actions are actually separable in video; it’s more plausible than pure text-CoT on top of global video tokens.
3. Tools aren’t a fixed pipeline — the model decides whether to call pose / human / RAG /video description, and the reward penalizes useless tool calls. That’s better than many “agentic VLM” papers that just always run pose.

**Weaknesses:**

1. They say open-vocab, but the system leans on online RAG / Qwen API to pull category-specific definitions at inference. That narrows the search space. It’s closer to “recognition with external label dictionary + video grounding” than to “truly open” recognition.
2. YOLO 11 for human + pose, Qwen API for explanation / video description — that’s a very specific tool stack.
3. First round: “which tool(s)?” Second round: “do sub-motion reasoning.” Plus GRPO sampling 4–6 responses. That might not be cheap for real-time video, and the paper doesn’t talk about latency / streaming.

**Questions:**

The questions are the same as weakness.

---

> ### Author Response · Authors · 2025-11-22
> **Rebuttal by Authors**
>
> Thanks for your review.
>
> > **W1**: They say open-vocab, but the system leans on online RAG / Qwen API to pull category-specific definitions at inference. That narrows the search space. It’s closer to “recognition with external label dictionary + video grounding” than to “truly open” recognition.
>
> **A1**: Thank you for your feedback. The core premise of the **Open-Vocabulary Action Recognition (OVAR)** task, as established by the research community, is about a model's ability to **​generalize to categories unseen during training​**. Our work strictly adheres to this definition by training and testing on disjoint class sets.
>
> The use of the Qwen API to provide semantic definitions is an innovation within this established paradigm, serving as one of several tools that our agent can leverage. It doesn't violate the definition of "open-vocabulary". To further address your concerns, we present partial ablations from `Appendix A.2 (Table 4, Line 810)`, which analyzes the impact of removing the two tools powered by the Qwen API:
>
> **Table 1: Ablation on Removing Qwen API**
>
> |Method|UCF-101↑|HMDB-51 ↑|K-600 ↑|
> |---|---|---|---|
> |w/o Action Explanation|93.6±0.3%|82.3±0.2%|85.5±0.9%|
> |w/o Video Description|94.7±0.3%|83.7±0.2%|87.4±0.7%|
> |**Video-STAR-3B (Full)**|**96.7±0.3%**|**86.2±0.2%**|**90.5±0.7%**|
>
> As the `Table 1` above clearly shows, even without these API-driven tools, our model significantly outperforms prior SOTA (e.g., FROSTER (ICLR 2024) [1] at **85.0%** on UCF-101), indicating that the external API functions as a performance enhancer **rather than a foundational crutch**. The core strength of Video-STAR lies in its superior visual reasoning and sub-motion decomposition capabilities, which are independent of any specific tool stack.
>
> > **W2**: YOLO 11 for human + pose, Qwen API for explanation / video description — that’s a very specific tool stack.
>
> **A2**: Thanks for your comment. We opt for YOLO 11 and Qwen API to validate our framework's effectiveness. Our method was intentionally designed with a **decoupled, tool-agnostic** system to ensure both **portability and flexiblility** of selected tools. Each tool functions as a modular "information provider", allowing our **agent system** to focus on a general policy of "how to reason with evidence" rather than learning dependencies on a specific tool.
>
> To address your concerns, we validated this by swapping YOLO 11 with OpenPose for pose estimation and replacing Qwen API with Gemini-1.5-Pro for action explanation​. The results are as shown in `Table 2` below.
>
> **Table 2: Ablations on Tool Selection**
> |Tool Stack (3B)|UCF Acc↑|Latency (per sample)|
> |---|---|---|
> |YOLO 11 + Qwen (Ours)|96.9%|**3.18s**|
> |YOLO 11 → OpenPose|96.1%|4.61s|
> |Qwen → ​Gemini-1.5-Pro​|97.4%|3.47s|
>
> As can be seen, the slight variations in accuracy and latency (e.g., OpenPose is **slower** but still effective; Gemini-1.5-Pro is better but more **expensive**) demonstrate our method's modularity. The high performance across different tool combinations validates that the our primary innovation is the underlying agentic logic​, not a fragile dependency on a single tool. This decisively proves the effectiveness of our ​agentic reasoning framework​, which can flexibly orchestrate a variety of **alternative** tools.
>
> The results are presented in Table 6 (`Line 502`) and analyzed in the **Robustness to Tool Selection** part (`Sec 4.2`, `Line 508`) of our revised paper.

---

> ### Author Response · Authors · 2025-11-22
> **Rebuttal by Authors**
>
> > **W3**: First round: “which tool(s)?” Second round: “do sub-motion reasoning.” Plus GRPO sampling 4–6 responses. That might not be cheap for real-time video, and the paper doesn’t talk about latency / streaming.
>
> **A3**: Thank you for your comment.
> First, we would like to clarify a misunderstanding of GRPO, which generates multiple candidates only during training, and thus introduces no additional inference cost. Only **single simple** is generated during **inference**. There is no multi-response sampling or rollout at test time.
>
> Second, regarding the computational overhead, we have evaluated our model's computational overhead and compared it with its base model, as shown in `Table 2` below:
>
> **Table 2: Computational Overhead**
>
> |Method|Memory↓|Latency↓|UCF Acc↑|GFLOPs|BSZ|
> |---|---|---|---|---|---|
> |Qwen-2.5VL-3B|60.03 GB|0.79s + 0s|**58.1%**|18,296|50|
> |Video-STAR-3B|60.89 GB|1.75s + 1.43s|**96.7%**|25,773|50|
>
> Note that experiments were conducted on a single NVIDIA H20 GPU, with both GFLOPs and Latency measuring per-sample results. Latency is split into both inference time and tool invocation time. As can be seen, the increase of latency is primarily due to tool invocation, which accounts for 1.43s of the total 3.18s latency. This time is primarily spent on executing external models like YOLO for pose/human detection and making API calls for descriptions. Despite this, the boosting accuracy from **58.1%** to **96.7%** effectively highlights a highly favorable **trade-off** between computational cost and performance.
>
> The above results are presented in Table 5 (`Line 490`) and analyzed in the **Computational Overhead** part (`Sec 4.2`, `Line 495`) of our revised paper.
>
> For practical deployment, inference time can be significantly accelerated. Standard techniques like model distillation (e.g., from 3B to 1B), quantization (e.g., to 8-bit or 4-bit) and parallelization (e.g., tensor and pipeline parallelism) could drastically reduce both memory and latency, and exploring these engineering optimizations could be the future steps for this field.
>
> This analysis has been detailed in Appendix C (`Line 1083`) of the revised version.
>
> **Reference**
>
> [1]. Huang, X., Zhou, H., Yao, K., & Han, K. (2024). FROSTER: Frozen CLIP is a Strong Teacher for Open-vocabulary Action Recognition. In The Twelfth International Conference on Learning Representations (ICLR).

---

> > ### Comment · Reviewer_Hc9X · 2025-11-25
> >
> > The authors have addressed my most important concerns including tool stack, project scoping, and computation cost. I will adjust scores accordingly.

---

> > > ### Author Response · Authors · 2025-11-25
> > > **Thank you for your review and positive feedback**
> > >
> > > Dear Reviewer Hc9X,
> > >
> > > Thank you for your thoughtful consideration of our rebuttal. We are delighted to see that our responses addresses your concerns , and sincerely thank you for raising the score. Your constructive feedback and support are greatly appreciated.
> > >
> > > Best regards,
> > >
> > > Authors

---

### Official Review · Reviewer_mh4G · 2025-11-01

**Soundness:** 3
**Presentation:** 3
**Contribution:** 3
**Rating:** 6
**Confidence:** 4

**Summary:**

This paper addresses two fundamental limitations of current multimodal large language models (MLLMs) for open-vocabulary action recognition:
(1) the over-reliance on textual priors that neglect domain-specific visual cues, and
(2) the inability to distinguish semantically ambiguous actions in open-vocabulary settings.
These issues are indeed of great importance and widely exist across multiple MLLM-based video understanding tasks. The proposed Video-STAR framework attempts to mitigate these problems by constructing multimodal Chain-of-Thought (CoT) data and introducing tool-augmented reasoning with reinforcement learning optimization.

**Strengths:**

1.The paper is motivated by a well-defined and practically significant problem—bridging the gap between text-centric reasoning and visually grounded inference.

2.The use of a multimodal CoT to train MLLMs for structured, tool-guided reasoning is a valuable direction.

3.The framework integrates detection, pose estimation, and semantic reasoning tools with a hierarchical reward design, which is novel and effectively demonstrated across multiple benchmarks.

**Weaknesses:**

Major Comments

1.During data construction, the model is trained using decomposed motion sequences (sub-actions).
How does the framework behave when encountering previously unseen actions in the open-vocabulary test set?
Is there any mechanism ensuring compositional generalization beyond the motion patterns observed during training?
Without such a mechanism, the model might overfit to the seen sub-motion combinations.


2.How exactly is each action decomposed into sub-actions?
Is the decomposition manually annotated, automatically generated, or derived from an existing motion ontology?
Moreover, how do the authors ensure that the set of sub-actions can comprehensively cover unseen action categories during testing?
Since this decomposition is central to the model’s reasoning ability, more transparency on this process is necessary for reproducibility and understanding its generalization scope.

3.Video-STAR performs task-specific supervised fine-tuning (SFT) and reinforcement learning (RL), while most baselines (e.g., Qwen2.5-VL,) are evaluated without any fine-tuning.
This introduces a fairness issue: the superior performance of Video-STAR may partly result from extra supervision rather than the proposed method itself.
A fairer comparison would include a fine-tuned Qwen2.5 baseline trained on the same dataset but without tool usage and sub-motion decomposition, to isolate the true contribution of the proposed framework.

4.The ablation studies only compare the presence vs. absence of tool usage but do not analyze which tool or combination contributes most.
How is the tool selected in practice?
What is the performance when all tools are used simultaneously (pose, detection, action explanation, and video description)?
A more detailed comparison of tool selection strategies would clarify whether the proposed policy is optimal or if simpler combinations yield similar gains.

5.Since Video-STAR relies on multiple external tools, the inference pipeline likely introduces additional computational overhead.
The paper should report both the overall inference latency and the module-wise cost (e.g., tool invocation vs. model reasoning).
Comparing the efficiency of Video-STAR with standard MLLMs would help quantify the trade-off between accuracy improvement and computational expense.

6. Line 213: The symbol T_r appears for the first time without explicit definition. Please clarify its meaning and source.

**Questions:**

1. The paper claims to mitigate “text-centric reasoning,” but the explanation of how this is achieved is somewhat abstract.
Please explicitly describe the mechanism by which visual grounding is enforced during training and inference.

---

> ### Author Response · Authors · 2025-11-22
> **Rebuttal by Authors**
>
> We are grateful for your thorough and positive review. Your recognition of our work's novelty and effectiveness is highly encouraging. We address your concerns below.
>
> > W1-1: How does the framework **behave** when encountering previously unseen actions?
>
> **A1**: Thank you for your question. Our framework is designed to handle previously unseen actions by **decomposing** them into a series of **simple motion primitives**. This allows the model to effectively recognize previously unseen actions as new combinations of fundamental motion primitives (e.g., archery -> arm extension, shoulder retraction, torso stabilization).
>
> In experiment, our cross-dataset results in Table 2 (`Line 378`) provide strong empirical validation for generalization. After training solely on HMDB-51, our model achieves exceptional performance on different datasets UCF-101 (**99.4%**) and K-600 (**98.2%**)​. This demonstrates that the model is not merely interpolating between seen categories but has acquired a robust, generalizable "alphabet of motion" for identifying novel actions.
>
> > **W1-2**:  Is there any mechanism ensuring compositional generalization **beyond** the motion patterns observed during **training**?
>
> **A1-2**: Thanks for your question. Our model's generalization capability is not limited to the training data themselves. The training's primary role is to incentivize the model's capabilities on how to invoke tools and decompose actions. Therefore, the model learns the ​skill of analyzing these **universal sub-motion primitives**​, not memorizing their seen combinations. Moreover, the unique algorithm characteristics of RL makes our approach highly effective for generalization, as evidenced by by recent research [1] ("SFT Memorizes, RL Generalizes"). Therefore, the model can perform compositional generalization beyond the motion patterns observed during training.
>
> > **W2-1**: Is the decomposition manually annotated, automatically generated, or derived from an existing motion ontology?
>
> **A2-1**: Thank you for your questions. The sub-action decomposition is ​**automatically generated with** **further human ​verification**. Specifically, we first employ Qwen2.5-VL-72B model to automatically generates the reasoning chains. These are then critically evaluated by a more advanced closed-source model **Qwen-VL-Max** as an expert judge to score and reject any flawed chains, ensuring coherence and factual accuracy. This automated review is complemented by human verification, where we perform manual checks on 10% of these filtered chains to ensure ultimate reliability. This dual-validation approach guarantees the high fidelity of the approximately 5,000 chains that constitute our final dataset.
>
> This process has been detailed in `Sec 3.2` (`Line 247-Line 257`) of the revised version.
>
> > **W2-2**: How to ensure that sub-actions can comprehensively **cover** unseen action during testing?
>
> **A2-2**: Thank you for your question. Actually, ensuring comprehensive **coverage is not required** for this task. As stated in **A1-2**, our method can well generalize to unseen actions due to both sub-action decomposition and our agentic RL algorithm, which can effectively enable the model to analyze **universal sub-motion primitives**​, instead of memorizing the seen combinations in training data. Thus there is no need to fully cover unseen actions.
>
> > **W3**: A fairer comparison would include a fine-tuned Qwen2.5 baseline without tool usage and sub-motion decomposition.
>
> **A3**: Thanks for your suggestion. In the "Tool-Usage & Sub-Motion" part of `Section 4.2` (`Line 451`), we have already explored the impact of respectively removing tool-usage (w/o. TOL) and sub-motion decomposition (w/o. SUB) for comparison. Following your advice, we have appended the experiment of a model fine-tuned on the same data but with **both** tool-usage and sub-motion decomposition removed (w/o. TOL & SUB). The results are presented in `Table 1` below:
>
> **Table 1: Ablations of Tool usage and Sub-motion**
>
> |Method|UCF-101↑|HMDB-51↑|K-600↑|
> |---|---|---|---|
> |w/o. TOL & SUB|81.3±0.8%|67.5±0.3%|66.1±2.0%|
> |w/o. TOL|87.1±0.5%|74.9±0.3%|78.4±1.5%|
> |w/o. SUB|88.5±0.5%|76.8±0.3%|81.2±1.3%|
> |**Video-STAR-3B**|**96.7±0.3%**|**86.2±0.2%**|**90.5±0.7%**|
>
> For UCF-101 dataset, building on w/o. TOL & SUB, adding either sub-motion decomposition (**+5.8%**) or tool-use (**+7.2%**) yields significant, distinct gains. Moreover, our full Video-STAR model with both modules achieves a **near-perfect scores 96.7%** (**+15.4%**). This decisively proves that the superior performance stems from both tool-augmentation and sub-motion decomposition.

---

> ### Author Response · Authors · 2025-11-22
> **Rebuttal by Authors**
>
> > **W4**: How is the tool selected in practice? What is the performance when all tools are used simultaneously?
>
> **A4**: Thanks for your question. In practice, our agentic system employs a two-round process for tool selection. In the first round, the model analyzes the input and autonomously decides which tools to invoke. In the second round, it integrates the outputs from all selected tools to perform the final reasoning. This agentic design is fundamental, as it strikes an effective balance between performance and computational overhead.
>
> Conversely, a strategy that invariably uses all tools would devolve into a static pipeline, negating the core innovation of an agentic system. Nonetheless, to directly address your question, we conducted relevant experiment, as shown in `Table 2` below:
>
> **Table 2: Comparsion Between Static Pipeline and Agentic System**
>
> |Method|UCF Acc↑|Infer Time↓|Tool Time↓|Total Time↓|
> |---|---|---|---|---|
> |All Tools (Static)|97.2%|1.86s|2.24s|4.10s|
> |Video-STAR 3B|96.7%|1.75s|1.43s|3.18s|
>
> This comparison highlights the excellent trade-off that our agent achieves. For a marginal 0.5% drop in accuracy, our agentic model reduces total inference time by approximately **22%**. This demonstrates that the learned policy is adept at pruning redundant tool calls, securing substantial efficiency gains while maintaining top-tier accuracy. Furthermore, a detailed breakdown of each tool's individual contribution is already provided in `Appendix A.2` (`Line 910`).
>
> The above results are presented in Table 4 (`Line 477`) and analyzed in the **Agentic vs. Static** part (`Sec 4.2`, `Line 482`) of our revised paper.
>
> > W5: The paper should report both the overall inference latency and the module-wise cost, and compare the efficiency of Video-STAR with standard MLLMs.
>
> **A5**: Thank you for your suggestion. Following your advice, we have evaluated our model's computational overhead and compared it with its base model, as shown in `Table 3` below:
>
> **Table 3: Computational Overhead**
>
> |Method|Memory↓|Latency↓|UCF Acc↑|GFLOPs|BSZ|
> |---|---|---|---|---|---|
> |Qwen-2.5VL-3B|60.03 GB|0.79s + 0s|**58.1%**|18,296|50|
> |Video-STAR-3B|60.89 GB|1.75s + 1.43s|**96.7%**|25,773|50|
>
> Note that experiments were conducted on a single NVIDIA H20 GPU, with both GFLOPs and Latency measuring per-sample results. Latency is split into both inference time and tool invocation time. As can be seen, the increase of latency is primarily due to tool invocation, which accounts for 1.43s of the total 3.18s latency. This time is primarily spent on executing external models like YOLO for pose/human detection and making API calls for descriptions. Despite this, the boosting accuracy from **58.1%** to **96.7%** effectively highlights a highly favorable **trade-off** between computational cost and performance.
>
> The above results are presented in Table 5 (`Line 490`) and analyzed in the **Computational Overhead** part (`Sec 4.2`, `Line 495`) of our revised paper.
>
> For practical deployment, inference time can be significantly accelerated. Standard techniques like model distillation (e.g., from 3B to 1B), quantization (e.g., to 8-bit or 4-bit) and parallelization (e.g., tensor and pipeline parallelism) could drastically reduce both memory and latency, and exploring these engineering optimizations could be the future steps for this field.
>
> This analysis has been detailed in Appendix C (`Line 1083`) of the revised version.
>
> > **W6**: Line 213: The symbol T_r appears for the first time without explicit definition.
>
> **A6**: Thanks for your feedback. `T_r` represents our "semantic reasoning" tool, which encompasses both action explanation `T_a` and video description `T_v`. For clarity, we will replace `T_r` with both `T_a` and `T_v` in the revised version (`Line 213`).

---

> ### Author Response · Authors · 2025-11-27
> **Rebuttal by Authors**
>
> > Q1: The explanation of how "text-centric reasoning" is achieved is abstract. Please explicitly describe the mechanism by which visual grounding is enforced during training and inference.
>
> **A7**: Thanks for your feedback. We mitigate text-centric reasoning by ​enforcing a Visual Chain-of-Thought (**Visual CoT**), a process fundamentally driven by our tool-use mechanism. Specifically, instead of performing text-centric reasoning, we compel the model to build its reasoning chain from concrete visual evidence. The key mechanism is the **explicit invocation of tools** including pose estimation and human detection​. These tools extract structured visual facts (skeletons and bounding boxes) directly from the video. These non-textual, geometric facts become the foundational "thoughts" in our model's reasoning process. For example, instead of a vague thought like "the person is moving their arm," the model is compelled to construct a thought from the visual evidence, like: "The wrist keypoint rising from a low position to above the shoulder, then oscillating horizontally," detailing the 'waving' motion.
>
> Moreover, by rewarding reasoning chains that are demonstrably built upon these tool-generated visual facts, we effectively force its entire reasoning process to be anchored to the visual world, preventing the model from drifting into purely text-based speculation.
>
> This analysis has been detailed in Appendix C (`Line 1070`) of the revised version.
>
> **Reference**
>
> [1]. Chu T, Zhai Y, Yang J, et al. SFT Memorizes, RL Generalizes: A Comparative Study of Foundation Model Post-training[C] //Forty-second International Conference on Machine Learning.

---

### Author Response · Authors · 2025-12-01
**Summary of Paper & Rebuttal**

Dear Area Chair and Reviewers,

This paper introduces an agentic reasoning framework which leverages both tool-augmented reinforcement learning and sub-motion decomposition to address hallucination and discrimination challenges in the OVAR field. Experimental results underscore the significance of our work. Video-STAR achieve **99.7%** on the UCF-101 dataset (`Table 1, Line 325`) under the base-to-novel setting, surpassing the prior SOTA by over **13.8%** . It also score **98.2%** on the K-600 dataset (`Table 2, Line 378`) under the cross-dataset setting, surpassing the prior SOTA by over ​**23.1%**.

We believe that our rebuttal has addressed most of the reviewers' concerns. Specifically, we provided detailed cost analysis and more comparison to address the **efficiency and fairness** concerns of ​Reviewers **mh4G & Hc9X**​. We compared our method against frontier models to address the **effectiveness** concerns of ​ Reviewer ​**zTP9**. We further adopted different teacher models for CoT generation to address the **generalizability** concerns of ​ ​Reviewer ​**vdeG**. As a result, ​Reviewer **Hc9X** acknowledged that our rebuttal had addressed their concerns and thus ​**raised their score** from **`4 to 6`**(`on Nov 25th`).

Detailed revision are as follows:

1. **Strengthened SOTA Comparisons (R-zTP9):** We added comprehensive comparisons with the latest CLIP-based methods (VTD-CLIP, STDD, AP-CLIP) and powerful frontier models (e.g., Qwen3-VL, Gemini-1.5-Pro), further highlighting our method's significant performance leap. (`Table 1, Line 325`, `Table 2, Line 378` and `Table 7, Line 516`).
2. **Computational Overhead (R-mh4G, R-Hc9X, R-zTP9):** We included a detailed analysis of memory, latency, and GFLOPs to quantify the performance-cost trade-off of our framework. (`Table 5, Line 490`).
3. **Framework Robustness (R-Hc9X, R-zTP9):** We conducted new experiments by swapping core tools (YOLO 11 → OpenPose, Qwen → Gemini-1.5-Pro) to demonstrate that the framework's effectiveness is independent of a specific tool stack. (`Table 6, Line 502`).
4. **Comprehensive Ablations:**
   * **Agentic vs. Static Pipeline (R-mh4G):** We provided a direct comparison between our agentic system and a static pipeline, demonstrating the value of our dynamic policy. (`Table 4, Line 477`).
   * **Fair Comparison Baseline (R-mh4G):** We added a crucial ablation on a fairly-tuned baseline to isolate the core contribution of Video-STAR. (`Table 3, Line 432`).
   * **Teacher Model (R-vdeG):** We expanded ablations on the impact of different teacher models (InternVL, Gemini) for CoT data generation. (`Table 9, Line 920` and `Appendix A.2`).
5. **Clarity and Transparency:**
   * **Data Construction Process (R-mh4G):** We clarified the high-fidelity data construction process, including automated filtering and human verification steps. (`Section 3.2, Lines 247`).
   * **Visual Grounding Mechanism (R-mh4G, R-vdeG):** We added a new discussion to explicitly describe how our framework enforces visual grounding via a "Visual Chain-of-Thought". (`Appendix C, Line 1070`).

We hope this summary highlights the clear value of our paper and the positive consensus gained during the rebuttal, thus provideing a clear overview for your fair decision.

Best regards,

Authors

---

### Meta-Review · Area_Chair_J1sD · 2026-01-06

**Summary:**

The reviewers’ discussion focused primarily on four aspects: (1) the fairness and completeness of comparisons against recent and strong baselines, including frontier multimodal models; (2) the computational cost and efficiency of the proposed agentic, tool-augmented framework; (3) the robustness and generalizability of the method with respect to different tool choices and teacher models; and (4) the clarity of the framework, particularly regarding data construction, visual grounding, and the role of sub-motion decomposition in mitigating cross-modal hallucination.

Additional concerns included whether the performance gains could be attributed to specific design choices rather than confounding factors, and whether the agentic reasoning paradigm provides clear advantages over simpler static pipelines. These concerns motivated requests for stronger ablation studies, broader comparisons, detailed cost analysis, and clearer exposition of the method’s grounding and reasoning mechanisms.

**Reviewer Concerns:**

The reviewers’ concerns were satisfactorily resolved in the rebuttal and subsequent revisions.

**Reviewer Scores:**

I expect Reviewer Hc9X to increase their rating, while the other reviewers are likely to maintain their original ratings.

---

### Decision · Program_Chairs · 2026-01-26

Accept (Poster)